# Hierarchical regulation of functionally antagonistic neuropeptides expressed in a single neuron pair

Ichiro Aoki [1,2] ✉, Luca Golinelli [3], Eva Dunkel[1,2], Shripriya Bhat[1], Erschad Bassam[1], Isabel Beets [3] & Alexander Gottschalk [1,2] ✉

Neuronal communication involves small-molecule transmitters, gap junctions, and neuropeptides. While neurons often express multiple neuropeptides, our understanding of the coordination of their actions and their mutual interactions remains limited. Here, we demonstrate that two neuropeptides, NLP-10 and FLP-1, released from the same interneuron pair, AVKL/R, exert antagonistic effects on locomotion speed in *Caenorhabditis elegans*. NLP-10 accelerates locomotion by activating the G protein-coupled receptor NPR-35 on premotor interneurons that promote forward movement. Notably, we establish that NLP-10 is crucial for the aversive response to mechanical and noxious light stimuli. Conversely, AVK-derived FLP-1 slows down locomotion by suppressing the secretion of NLP-10 from AVK, through autocrine feedback via activation of its receptor DMSR-7 in AVK neurons. Our findings suggest that peptidergic autocrine motifs, exemplified by the interaction between NLP-10 and FLP-1, might represent a widespread mechanism in nervous systems across species. These mutual functional interactions among peptidergic co-transmitters could fine-tune brain activity.

Neuropeptides can establish orthogonal signaling networks with important effects on behavior, operating either in conjugation with, or independently of, classical small-molecule or monoamine transmitters and gap junctions[1–3]. While neuropeptides can signal locally to post-synaptic or neighboring neurons[4–6], their capability extends to the global relay of information, received by numerous cells throughout the brain and body, as exemplified by peptide hormones like oxytocin, vasopressin, insulin, and pituitary-derived peptides[7]. The versatility of neuropeptides is evident in their involvement in various disorders: For instance, individuals lacking pro-opiomelanocortin (POMC) suffer from severe obesity[8], dysfunction of the orexin system can lead to narcolepsy[9], and alterations in the level of the neuroprotective neuropeptide Y (NPY) are observed in neurodegenerative disorders[10].

Frequently, multiple (neuro)peptides, occasionally with diverse or opposing functions, are expressed in the same cell[11–13]. However, it remains controversial whether distinct peptides are co-packaged into the same dense core vesicles (DCVs)[14–16] and whether the general excitation of these neurons leads to the universal release of all peptides. Some cases of differential packaging or regulation of multiple neuropeptides expressed in the same cell have been reported: For example, murine chromaffin cells have two populations of DCVs[17]. In pituitary neurons, POMC may be secreted from the constitutive pathway, while its derivative, adrenocorticotropic hormone (ACTH), is released from the regulated pathway[18–20]. Vasopressin and galanin are also packaged into different DCV populations[21]. Bag cells and the atrial gland of *Aplysia californica* contain DCVs with distinct sets of neuropeptides, and even multiple peptides from the same propeptide are distributed across different vesicles[22–25]. To facilitate such segregation, at least the initial cleavage of propeptides has to occur in the Golgi apparatus, before their loading into secretory vesicles, as suggested in

[1]Buchmann Institute for Molecular Life Sciences, Goethe University, Max-von-Laue-Strasse 15, D-60438 Frankfurt, Germany. [2]Department of Biochemistry, Chemistry and Pharmacy, Institute of Biophysical Chemistry, Goethe University, Frankfurt, Germany. [3]Department of Biology, KU Leuven, Leuven, Belgium. ✉e-mail: Aoki@bio.uni-frankfurt.de; a.gottschalk@em.uni-frankfurt.de

the processing of POMC[20]. In *C. elegans*, the release of multiple neuropeptides from ASI sensory neurons could be differentially regulated by monoamines[26], and oxidative stress increases the secretion of FLP-1 but not FLP-18 neuropeptides from AIY interneurons[27]. Co-transmission with multiple transmitters generally provides flexibility to neuronal microcircuits. This can happen either by convergently regulating the same target neuron conferring antagonistic, additive, or synergistic effects, or by divergently regulating multiple target neurons. Another strategy for flexibility involves packaging different transmitters into distinct vesicles and regulating their release specifically, providing an additional layer of adaptability[11]. Despite these observations, little is known about whether and how different neuropeptides in the same neuron might be differentially regulated.

To address this issue, we studied the AVK interneurons in *C. elegans*. This neuron pair stands out for its expression of a diverse array of neuropeptides and neuropeptide receptors, making it a central hub in the neuropeptide network of *C. elegans*[1]. Despite the absence of known small-molecule transmitters[28,29], AVK neurons signal to numerous cells in the nervous system, exerting influence over locomotion, roaming strategies, egg-laying[30-32], and pathogen avoidance[33]. They integrate various sensory inputs, including information about food presence[30], oxygen levels[31], and proprioceptive information[32], through a range of chemical and electrical synapses, including dopamine signaling[34-36]. Some of the neuropeptides expressed in AVK exhibit antagonistic effects on multiple aspects of locomotion. The most abundantly expressed neuropeptide in AVK, at the mRNA level, is FLP-1[37]. Since *flp-1* mutants are hyperactive, FLP-1 is proposed to suppress locomotion speed[38,39]. In addition, *flp-1* mutants show increased body bending angles[30-32,38,39]. Bending angles correlate with locomotion strategy, thus altering bending angles can affect the directionality and efficiency of locomotion, enabling the optimization of tactic behaviors[30,31,40]. When overexpressed, NLP-49, the second-most abundant neuropeptide in AVK, enhances the increase of locomotion speed in response to mechanical stimuli[41]. This suggests that FLP-1 and NLP-49 may exert antagonistic effects on locomotion speed, although the underlying mechanisms remain unclear.

Here, we demonstrate the antagonistic relationship between FLP-1 and NLP-10, two neuropeptides expressed in AVK. While FLP-1 released from AVK reduced locomotion speed, optogenetic depolarization of AVK unexpectedly accelerated locomotion. Through an AVK-specific RNAi-screen for neuropeptides, we found that NLP-10 mediates the increase of locomotion speed evoked by AVK-photoactivation. Furthermore, NLP-10 was required for the physiological escape response to mechanical stimuli and noxious blue light. Consistent with the functional antagonism, AVK-derived FLP-1 suppressed NLP-10 release from AVK through autocrine feedback. NLP-10 affected AVB and AIY interneurons, known to activate forward locomotion, via the excitatory G protein-coupled receptor (GPCR) NPR-35. Our findings uncover a "wireless" microcircuit governing locomotion speed and body posture, unpredicted by the anatomical connectome. Furthermore, we demonstrate hierarchical regulation between functionally antagonistic neuropeptides secreted from the same neuron.

## Results

### FLP-1 released from AVK neurons suppresses excitability in a modality-specific manner

*flp-1* mutant animals are hyperactive[38,39]. Although the FLP-1 neuropeptide is mainly produced in AVK[37], it appears to be synthesized and functional also in other neurons[27,38]. Thus, the source of FLP-1 affecting locomotion speed remained unclear. To address this, we employed the Multi Worm Tracker (MWT, Fig. 1a)[42], a device allowing to monitor locomotion of *C. elegans*. In the presence of food bacteria, consistent with previous reports[38,39], *flp-1* mutants displayed elevated basal locomotion speed compared to wild type animals (Fig. 1a–c and Supplementary Fig. 1a, f, g). In response to mechanical tapping stimuli on

the culture dish, wild type animals exhibited a speed increase followed by a gradual return to basal speed over the next 5 minutes[41-45]. In contrast, *flp-1* mutants showed an enhanced speed increase compared to wild type (Fig. 1a, d and Supplementary Fig. 1f, h). The body posture was also affected by the *flp-1* mutation, as evidenced by increased body bending angles in *flp-1* mutants (Supplementary Fig. 1b, c, i, j). Upon tapping, wild type animals decreased body bending angles, while *flp-1* mutants increased them (Fig. 1e, g and Supplementary Fig. 1i, k). To assess locomotion directionality, we calculated 'straightness' (Supplementary Fig. 1e and see methods). After tapping, straightness was lower in *flp-1* mutants, reflecting their "loopier" locomotion (Fig. 1h–j), which correlates with their increased bending angle, and can be recognized in the animal trajectories (Fig. 1a). Most defects observed in *flp-1* mutants were rescued by selectively expressing FLP-1 in AVK, using either the *twk-47* promoter[46] or the truncated *flp-1* promoter[30] (Fig. 1b–j and Supplementary Fig. 1f–k). This suggests that FLP-1 released from AVK slows down basal locomotion, suppresses the speed increase upon tapping, decreases body bending and straightens locomotion direction. In a complementary experiment, we induced an AVK-specific knockdown of *flp-1* by feeding dsRNA-producing bacteria (Fig. 1k–p). To achieve AVK-specific RNA interference (RNAi), we employed animals expressing the RNA endonuclease RDE-1 and the RNA transmembrane transporter SID-1 in AVK of RNAi-deficient *rde-1* mutants[47]. Consistently, AVK-specific *flp-1* knockdown enhanced the speed increase upon tapping. These results suggest that AVK-derived FLP-1 is required and sufficient to suppress (excessive) excitability, thereby mitigating hyperactivity.

To assess the general impact of FLP-1 on locomotion speed, we subjected animals to another type of acute sensory stimulation. Exposure to noxious blue light accelerates locomotion, a process primarily reliant on the light sensor protein LITE-1[48-50]. Unlike the response to tapping, the *flp-1* mutation did not enhance the speed increase upon blue light illumination (Supplementary Fig. 1l–q). This implies that FLP-1 modulates locomotion in a modality-dependent manner, rather than exerting a general slowing effect, which would be unfavorable in the case of escape behavior evoked by noxious stimuli.

Previously, the neuropeptide NLP-49 was implicated in the regulation of locomotion speed, since its overexpression in AVK enhanced the speed increase in response to mechanical stimulation[41]. However, mutants carrying the *nlp-49(gk546875)* nonsense allele did not exhibit any obvious alteration in locomotion speed[41], prompting further consideration of whether NLP-49 genuinely regulates locomotion speed. We, therefore, created additional deletion alleles of *nlp-49* and examined their effects on locomotion as well as responses to tapping and blue light. However, NLP-49 appeared not to exert an impact on locomotion speed during the tested scenarios, despite its abundant expression in AVK (Supplementary Fig. 2). Thus, we did not further investigate NLP-49.

### AVK photoactivation accelerates directional locomotion, coordinated by AVK-derived FLP-1

Previously, we demonstrated that acute manipulation of AVK activity off food, using Halorhodopsin (NpHR) or Channelrhodopsin-2 (ChR2), increased or decreased body bending, respectively, in line with the role of tonic FLP-1 release from AVK in reducing body bending angles[30]. To probe the influence of AVK on locomotion speed without inducing photophobic behavior by the blue light used for ChR2 activation, we selectively expressed the red-light activated ChR Chrimson[51] in AVK using the *twk-47* promoter (Supplementary Fig. 3a). In the absence of the ChR chromophore all-*trans* retinal (ATR), red light had no impact on locomotion (Supplementary Fig. 3b). However, with ATR, despite AVK-derived FLP-1 slowing down locomotion (Fig. 1), and despite far higher expression of *flp-1* mRNA compared to other neuropeptide mRNAs in AVK[30,37,46], AVK-photoactivation slightly accelerated locomotion (Fig. 2a–c), consistent with AVK activity correlating with

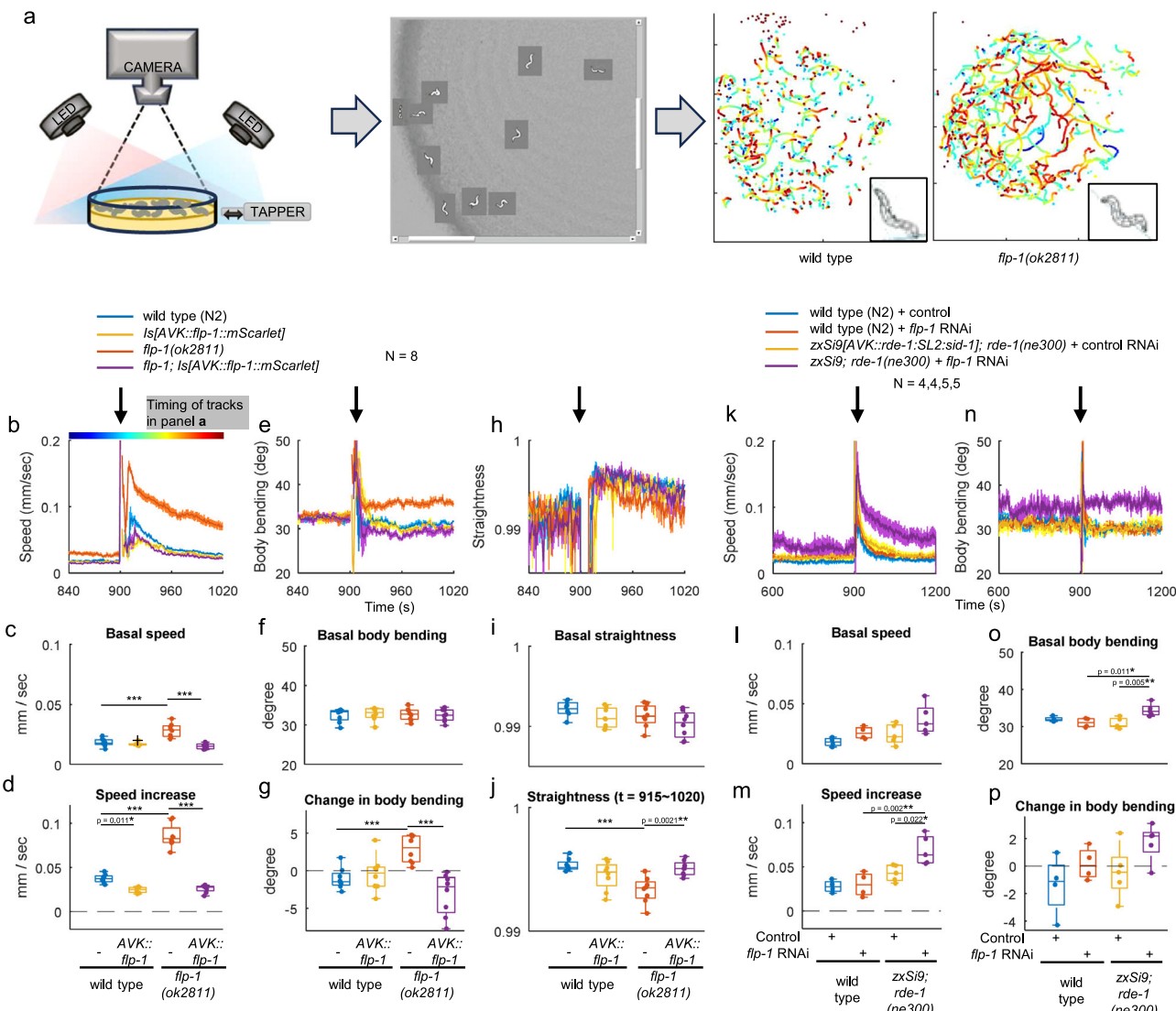

**Fig. 1 | AVK-derived FLP-1 suppresses excessive excitability. a** Multi-Worm Tracker (MWT) setup with blue and red illumination and tapping device[42]. Representative trajectories obtained for wild type and *flp-1* mutant animals are shown. Color code represents time as shown in **b**i. Representative animal skeletons after tapping (t = 920) are shown in insets. Note that enhanced and prolonged 'speed increase' after tapping of *flp-1* mutants are represented by longer turquoise-red traces. **b–j** Locomotion of animals with the indicated genotype was tracked using MWT. NGM plates were tapped three times at 1 Hz, 900 s after recording began. 'Body bending' and 'straightness' were defined as described in Supplementary Fig. 1e. Basal speed (**c**), body bending (**f**) and straightness (**i**) indicate mean values from t = 840 to 899. 'Speed increase' (**d**) and 'Change in body bending' (**g**) indicate mean values from t = 911 to 960 subtracted by basal values. The mean straightness after tapping (t = 911 - 1020) was plotted in **j**. Tukey test was performed.

***$p < 0.001$. Note that tracks are temporally lost upon tapping, causing large variations and losses of data points. Approximately 20–120 animals were involved in each recording. **k–p**. Wild type N2 strain and animals for AVK-specific feeding RNAi strain (*zxSi9[ptwk-47::rde-1:SL2:sid-1]; rde-1(ne300)*) were fed with HT115 bacteria carrying a control vector or a plasmid encoding RNAi for *flp-1*. Animals were then subjected to behavioral analysis as described in **a**. Tukey test was performed. Approximately 20-80 animals were involved in each recording. N indicates the numbers of independent experiments from different populations. In timeseries plots, data are presented as mean values +/- standard error of the mean (SEM). In boxplots, the boxes extend from the first quartile (Q1) to third quartile (Q3), with the band inside the boxes representing the median. The whiskers extend to the smallest and largest values within 1.5 times the inter-quartile range (IQR), where IQR is the difference between Q3 and Q1. Source data are provided in 'Source Data 1' file.

locomotion speed[31]. In *flp-1(ok2811)* mutants, the speed increase upon AVK-photoactivation was enhanced (Fig. 2a–c). AVK photoactivation decreased body bending in wild type but conversely increased it in *flp-1* mutants, thus anti-correlating the straightness of locomotion (Fig. 2d–f and Supplementary Fig. 3c). This is in accordance with the established role of FLP-1 in reducing body bending. Thus, AVK (photo) activation causes faster and straighter locomotion, that could facilitate efficient dispersal or escape behavior, while without FLP-1, signaling from AVK causes even faster but less directional locomotion. The *flp-1* mutation also enhanced speed increase of animals expressing

ChR2(C128S), a ChR2 mutant activatable at lower light intensities[52], in AVK by the truncated *flp-1(trc)* promoter, affirming that FLP-1 suppresses the speed increase induced by AVK photoactivation (Supplementary Fig. 3d). Moreover, AVK-specific feeding RNAi for *flp-1* enhanced the speed increase and reversed the change in body bending from decrement to increment (Fig. 2g–l). In sum, these results indicate that a signal originating from AVK, distinct from FLP-1 and induced by cell depolarization, accelerates locomotion. Additionally, they highlight the role of FLP-1 released from AVK in decelerating and straightening locomotion.

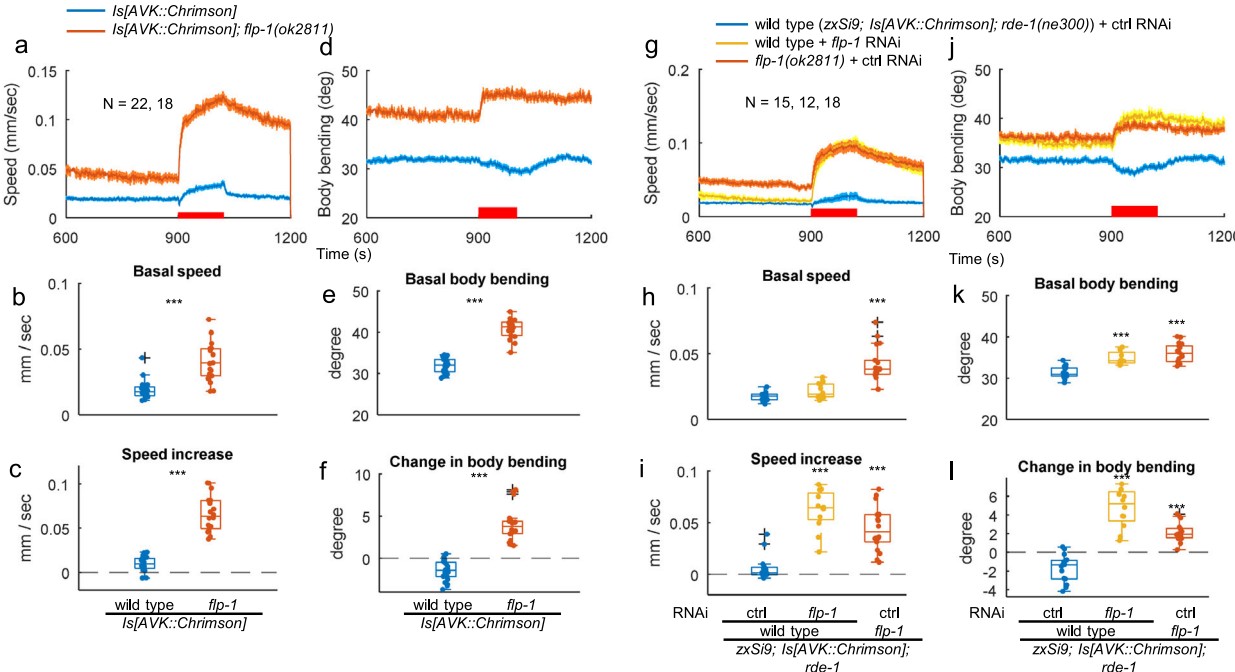

**Fig. 2 | AVK photoactivation accelerates locomotion, which is suppressed by AVK-derived FLP-1. a–f.** Wild type and *flp-1(ok2811)* mutant animals expressing Chrimson specifically in AVK were cultured on NGM plates supplemented with ATR and subjected to behavioral analysis with continuous red light illuminated from t = 900 to 1020 to activate Chrimson. Basal values are the mean from t = 780 to 899. 'Speed increase' (**c**) and 'Change in body bending' (**f**) indicate mean values from t = 900 to 1020 subtracted by basal values. Two-tailed Welch test was performed. ***p < 0.001. Approximately 20–120 animals were involved in each recording. Part of measurements are overlapping with Supplementary Fig. 3e. **g–l**. Wild type and *flp-1* mutant RNAi strains expressing Chrimson in AVK were cultivated with HT115 bacteria carrying a control vector or a plasmid producing dsRNA for *flp-1* with ATR for 3 days and subjected to behavioral analysis. Dunnett test was performed against wild type animals fed with control HT115 bacteria. ***p < 0.001. Approximately 25–120 animals were involved in each recording. In timeseries plots, data are presented as mean values +/- SEM. In boxplots, the boxes extend from the Q1 to Q3, with the band inside the boxes representing the median. The whiskers extend to the smallest and largest values within 1.5 times the IQR. Source data are provided in 'Source Data 1' file.

## NLP-10 mediates the AVK-derived speed increase

We asked whether chemical or electrical transmission from AVK may be responsible for the observed enhanced speed increase in response to photoactivation. Initially, we probed electrical transmission. AVK expresses gap junction (GJ) subunits such as UNC-7, INX-7 and INX-19[37,53], among others. However, RNAi knockdown of these GJ subunits, along with others, did not suppress the speed increase of *flp-1* mutants following AVK photoactivation (Supplementary Fig. 4b). Similarly, a dominant-negative variant of UNC-1, a stomatin protein required for function of GJs containing UNC-9[54,55], did not impede the speed increase (Supplementary Fig. 4a). Thus, GJs do not appear to be involved in the speed increase following AVK photoactivation.

Next, we examined the general role of chemical transmission. Tetanus toxin light chain (TeTx) blocks exocytosis both from synaptic vesicles and DCVs through cleavage of synaptobrevins[56,57]. Expression of TeTx partially suppressed the increases in speed and body bending upon AVK photoactivation in *flp-1* mutants (Fig. 3a–c and Supplementary Fig. 5a–c), suggesting that exocytosis is at least partially responsible for the effect of AVK photoactivation. Although no small-molecule transmitter is known to be released by AVK[28,29], and most of the genes involved in biogenesis and transmission of acetylcholine (ACh), glutamate, GABA and monoamines are not expressed, trace amounts of *eat-4* and *cat-1*, which encode vesicular transporters for glutamate and monoamines, respectively, might be expressed in AVK[37]. However, RNAi for these did not suppress the speed increase (Supplementary Fig. 4c).

Hence, we investigated if a neuropeptide is responsible for the observed speed increase. To this end, 19 neuropeptide genes expressed in AVK besides FLP-1[37] were knocked down by AVK-specific feeding RNAi. Among these, only the knockdown of *nlp-10* significantly attenuated the speed increase in *flp-1* mutant animals (Supplementary Fig. 4d and Fig. 3d–f). To further validate the role of NLP-10, we generated two *nlp-10* alleles, *zx28* and *zx29*, using CRISPR/Cas9-mediated genome editing. Consistent with the RNAi result, these alleles suppressed the speed increase upon AVK photoactivation in *flp-1(ok2811)* mutants (Fig. 3g–i), solidifying the involvement of NLP-10. The extent of suppression was consistent with the nature of the mutations (Fig. 3g–i): The *zx28* allele might still produce one of four NLP-10 peptides (NLP-10-4), while the *zx29* allele is a putative null (Supplementary Fig. 4e). The reduced straightness in *flp-1* mutants was suppressed by the *nlp-10* mutation (Supplementary Fig. 5q), suggesting that the directionality of the locomotion is also antagonistically regulated by FLP-1 and NLP-10. The *nlp-10(tm6232)* allele, which deletes only the second exon, did not suppress the speed increase, likely due to the residual coding regions and intact splice sites allowing the generation of all four mature peptides (Supplementary Fig. 4e, f). In summary, our findings suggest that NLP-10 neuropeptides underlie the AVK-induced speed increase.

NLP-10 overexpression in AVK of wild type animals enhanced speed increase upon AVK photoactivation and reversed the effect of AVK photoactivation on body bending from negative to positive (Fig. 3l and Supplementary Fig. 5l). The expression and secretion of NLP-10 was confirmed by visualizing mScarlet fused to the NLP-10 prepropeptide in scavenger cells (coelomocytes) that endocytose and filter the body fluid (Supplementary Fig. 4g)[58]. The *flp-1* mutation increased body bending of animals expressing NLP-10 both before and during AVK photoactivation (Supplementary Fig. 5j, m), suggesting that FLP-1 has additional functions independent of NLP-10. Overexpression of NLP-10 in other neurons that natively express *nlp-10*,

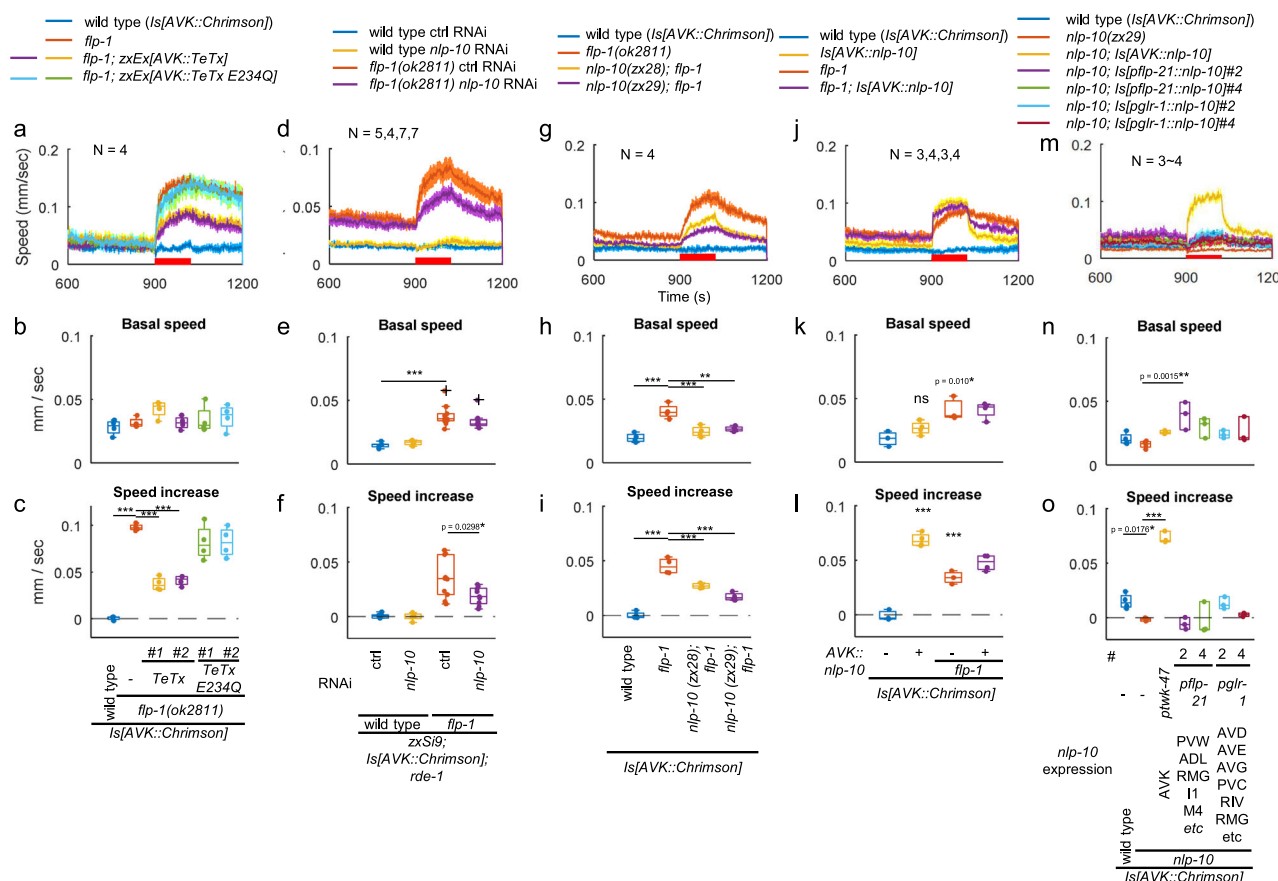

**Fig. 3 | NLP-10 is the AVK-derived accelerator. a–c** L4 larvae of wild type and *flp-1(ok2811)* mutant animals, along with *flp-1* mutants expressing either wild type or enzymatically inactive (E234Q) TeTx in AVK, all expressing Chrimson in AVK, were picked and transferred to NGM plates with ATR, cultivated overnight and subjected to behavioral analysis with MWT while red light was illuminated as indicated. **a** Speed along time, **b** average speed before tapping (t = 780 - 899, $b_o$) and (**c**) speed increase during illumination (average speed (t = 900 - 1020) − $b_o$) were plotted. Dunnett test was performed against *flp-1* mutants. ***$p < 0.001$. Approximately 10–50 animals were involved in each recording. **d–f.** Wild type and *flp-1(ok2811)* mutant RNAi strains expressing Chrimson in AVK were allowed to self-fertilize on NGM plates with HT115 bacteria carrying a control vector or a plasmid producing dsRNA for *nlp-10* in the presence of ATR for 3 days and subjected to behavioral analysis. Tukey test was performed. ***$p < 0.001$. Approximately 30–100 animals were involved in each recording. **g–i.** Animals expressing Chrimson in AVK

with indicated genotypes were analyzed. Tukey test was performed. ***$p < 0.001$. Approximately 20–100 animals were involved in each recording. **j–l.** Wild type and *flp-1(ok2811)* mutant animals expressing Chrimson with or without AVK-specific NLP-10 overexpression from *zxIs171[ptwk-47::nlp-10]* were analyzed. Tukey test was performed. ***$p < 0.001$. Approximately 50-150 animals were involved in each recording. **m–o.** *nlp-10(zx29)* mutants expressing NLP-10 in indicated neurons, along with wild type and *nlp-10(zx29)* mutant animals, all expressing Chrimson in AVK, were analyzed. Dunnett test was performed against *nlp-10* mutants. ***$p < 0.001$. Approximately 50–120 animals were involved in each recording. Chrimson was expressed in AVK from *zxIs153[ptwk-47::Chrimson]*. In timeseries plots, data are presented as mean values +/- SEM. In boxplots, the boxes extend from the Q1 to Q3, with the band inside the boxes representing the median. The whiskers extend to the smallest and largest values within 1.5 times the IQR. Source data are provided in 'Source Data 1' file.

using the *flp-21* and *glr-1* promoters, did not induce an increase of locomotion speed or body bending upon AVK photoactivation (Fig. 3m–o and Supplementary Figs. 5n–p and 6a; we confirmed expression and secretion of NLP-10, see Supplementary Fig. 4h, i). This suggests that systemically enriched NLP-10 is not sufficient to induce the speed increase upon AVK-photoactivation, emphasizing the necessity of spatially and/or temporally specific release of NLP-10 from AVK.

### AVK-derived FLP-1 reduces NLP-10 secretion

We investigated the mechanistic basis of how FLP-1 may suppress the NLP-10-mediated speed increase. As a premise, we confirmed that AVK excitability upon photoactivation was not increased in *flp-1* mutants (Supplementary Fig. 7a). FLP-1 could suppress the speed increase either 1) through AVK-intrinsic effects on the expression or release of NLP-10; 2) through effects on downstream circuitry, e.g. having antagonistic effects on the same target neuron, or influencing different neurons with similar or antagonistic functions. *nlp-10* transcription in

AVK was not increased in *flp-1* mutants (Supplementary Fig. 7b), suggesting that FLP-1 does not suppress NLP-10 by decreasing its expression. To address whether FLP-1 affects secretion of NLP-10 from AVK, we quantified fluorescence of mScarlet co-secreted with NLP-10 and taken up by coelomocytes. The normalized mScarlet fluorescence in coelomocytes was largely increased in *flp-1* mutants (Fig. 4a and Supplementary Fig. 7c, d). In the *flp-1* mutant background, additional mutation of *unc-31(n1304)*, which encodes the CAPS protein responsible for exocytosis from DCVs[59], reduced mScarlet fluorescence in coelomocytes (Supplementary Fig. 7d). This indicates that release of NLP-10 from AVK is dependent of UNC-31 as was the case for FLP-1[30], and suggests that in *unc-31* single mutants, release of NLP-10 is affected both by loss of *unc-31* and as a result of the decreased FLP-1 release. AVK-specific knockdown of *flp-1* by RNAi also increased NLP-10 release from AVK (Fig. 4b). Taken together, although the possibility of FLP-1 effects on downstream circuits is not excluded, our results align with the idea that AVK-derived FLP-1 can suppress NLP-10 secretion from AVK.

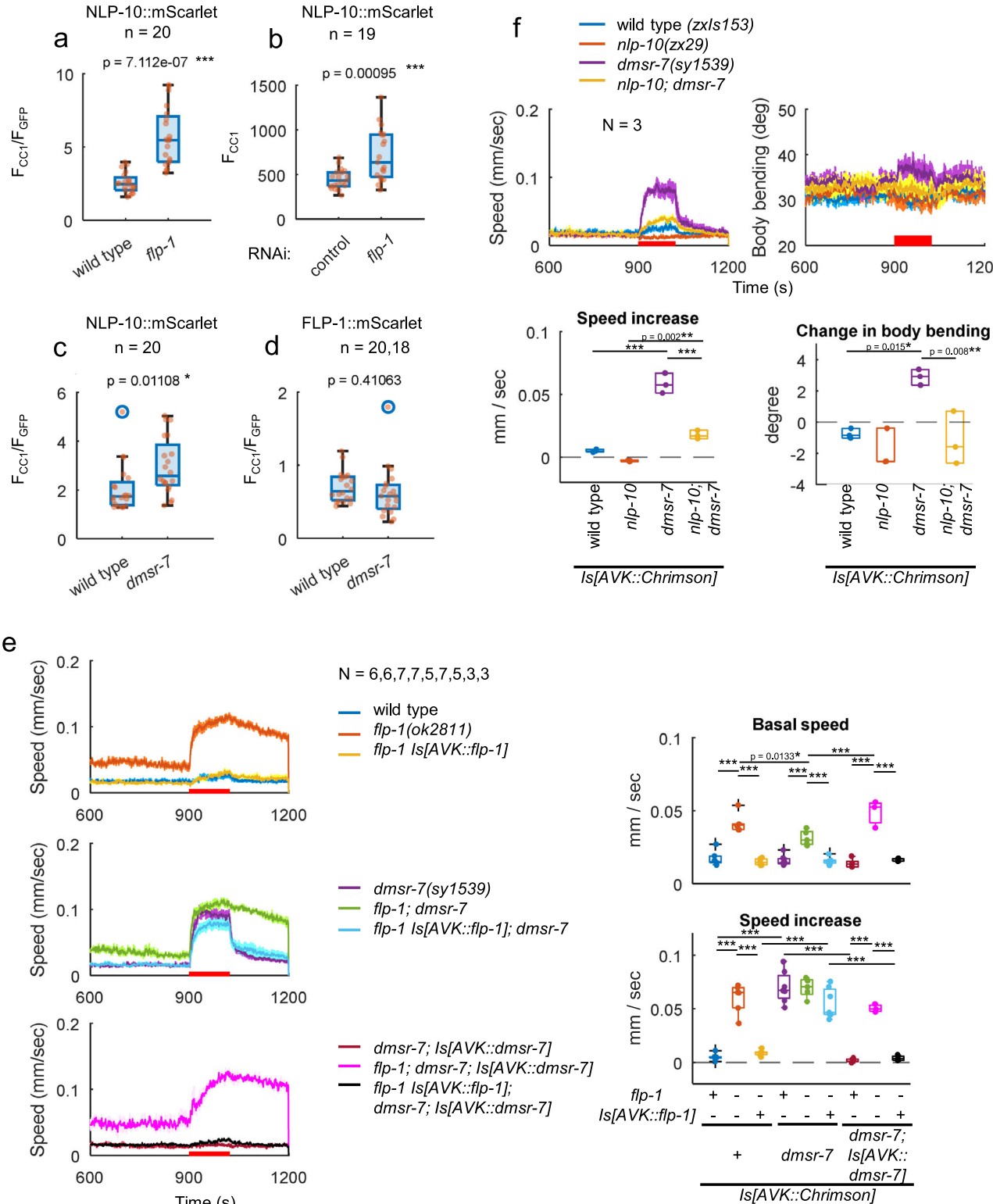

**Fig. 4 | AVK-derived FLP-1 suppresses NLP-10 secretion from AVK through autocrine feedback.** **a** Fluorescence intensities of mScarlet in L4 larvae of wild type and *flp-1(ok2811)* mutant animals expressing NLP-10::mScarlet in AVK were measured in anterior coelomocytes ($F_{CC1}$). Fluorescence intensities of GFP expressed in AVK under the same promoter as NLP-10::mScarlet were also measured ($F_{GFP}$). The fluorescence intensities of mScarlet in coelomocytes divided by that of GFP in AVK ($F_{CC1}/F_{GFP}$) were plotted. **b** AVK-specific feeding RNAi strain expressing NLP-10::mScarlet in AVK was cultivated with HT115 bacteria carrying a control vector or a plasmid producing dsRNA for *flp-1* for 3 days, and $F_{CC1}$ from L4 larvae were plotted. Wild type and *dmsr-7(sy1539)* mutant animals expressing NLP-10::mScarlet (**c**) or FLP-1::mScarlet (**d**) in AVK were analyzed as described in (**a**).

Two-tailed Welch test was performed. **e, f.** Animals of indicated genotypes expressing Chrimson in AVK were cultivated with ATR and subjected to behavioral analysis with MWT while red light was illuminated as indicated. Tukey test was performed. ***$p < 0.001$. 40–120 animals were involved in each measurement. Part of wild type and *dmsr-7* data are overlapping between **e** and **f**. Approximately 20–120 animals were involved in each recording. In timeseries plots, data are presented as mean values +/- SEM. In boxplots, the boxes extend from the Q1 to Q3, with the band inside the boxes representing the median. The whiskers extend to the smallest and largest values within 1.5 times the IQR. Source data are provided in 'Source Data 1' file.

We next addressed whether and how AVK activation acutely affects the release of FLP-1 and NLP-10. AVK photoactivation in animals overexpressing NLP-10 in AVK accelerated locomotion speed (Fig. 3j–o). However, when photoactivation was repeated with pulsed illumination, the response of animals gradually decreased (Supplementary Fig. 7g). Meanwhile, the release of FLP-1 and NLP-10 increased and decreased, respectively. These results suggest that when AVK is intermittently activated, increased FLP-1 release suppresses NLP-10 release, leading to suppression of speed increase.

### The FLP-1 receptor DMSR-7 acts in AVK to mediate autocrine feedback

How might FLP-1, released from AVK, suppress NLP-10 secretion from AVK? We addressed two hypotheses: 1) FLP-1 and NLP-10 compete for intracellular machinery during their biosynthesis or packaging into DCVs within AVK, or 2) FLP-1 secreted from AVK affects AVK itself by autocrine or circuitry feedback, thereby suppressing the release of NLP-10. Expression of truncated FLP-1 propeptides, comprising three or fewer of the N-terminal mature peptides, failed to rescue defects of the *flp-1* mutants, although their secretion was confirmed by coelomocyte uptake of co-released mScarlet (Supplementary Fig. 8a). Thus, functional rescue seems to require more C-terminal FLP-1 peptides. Since the NLP-10 function was not suppressed even when excessive amounts of N-terminal FLP-1 species were expressed and released, it seems unlikely that FLP-1 suppresses NLP-10 release by competing for intracellular machinery.

We next addressed the possibility that AVK-derived FLP-1 affects AVK through autocrine feedback. To this end, there should be a receptor of FLP-1 that is expressed in AVK and relevant in this context. Various receptors for FLP-1 have been described in vivo and in vitro, such as NPR-6, FRPR-7[30], NPR-4[27,60], NPR-5[61], NPR-11[62], NPR-22[63,64], DMSR-1, DMSR-5, DMSR-6 and DMSR-7[2]. Among these, *dmsr-6 and dmsr-7* are expressed in AVK neurons (Supplementary Fig. 6b)[37]. *dmsr-7(sy1539)* loss-of-function mutant animals exhibited an enhanced speed increase and increased body bending upon AVK photoactivation, similar to *flp-1* mutants (Fig. 4e and Supplementary Fig. 8b). However, unlike *flp-1* mutants, *dmsr-7* mutants did not show an increase in basal speed or body bending, nor did they exhibit a prolonged speed increase after photostimulation, indicating that these effects of FLP-1 are possibly mediated by other receptor(s) (Fig. 4e and Supplementary Fig. 8b). *flp-1; dmsr-7* double mutants did not display an additive defect compared to *flp-1* single mutants. In addition, while the defects of *flp-1* mutants were mostly rescued by FLP-1 expression in AVK, this rescue effect was largely compromised by loss of *dmsr-7* (Fig. 4e). The defects caused by the *dmsr-7* mutation were rescued by expressing DMSR-7 specifically in AVK (Fig. 4e). The enhanced AVK-evoked speed increase of *dmsr-7* mutants was suppressed by the *nlp-10* mutation (Fig. 4f), as was the case for *flp-1* mutants (Fig. 3g–i), suggesting that NLP-10 acts downstream of FLP-1/DMSR-7 signaling in AVK. The secretion of NLP-10 from AVK was increased in *dmsr-7* mutants, similar to *flp-1* mutants (Fig. 4c), consistent with reported inhibitory effects of DMSR-7 through $G\alpha_{i/o}$-coupling[2,33,65]. By contrast, secretion of FLP-1 was not increased by *dmsr-7* mutation (Fig. 4d and Supplementary Fig. 7f), suggesting that the secretion of NLP-10 and FLP-1 is separately regulated. *dmsr-7* is expressed in other neurons such as AVB and VB. Expression of *dmsr-7* in AVB partially rescued the defect of *dmsr-7* mutants, which was dependent on *flp-1* (Supplementary Fig. 8c, d). This suggests that FLP-1/DMSR-7 signaling has a secondary pathway that directly suppresses premotor AVB neurons, thereby counteracting the speed increase induced by AVK-photoactivation. In sum, these results suggest that FLP-1/DMSR-7 signaling primarily reduces NLP-10 secretion from AVK via autocrine feedback, resulting in suppression of the AVK-evoked speed increase and reduction of body bending, which correlates with straightened locomotion.

### NPR-35 functions downstream of NLP-10 as its receptor

We previously identified NPR-35 as a receptor of NLP-10 in vitro[2]. NPR-35 belongs to the neuropeptide FF (NPFF) and SIFamide receptor family, a conserved family across bilaterian animals, including vertebrate NPFF receptors and protostomian SIFamide receptors[2,66,67] (Supplementary Fig. 9). Mammalian NPFF is involved in various behaviors such as nociception[68]. Therefore, we examined the requirement of NPR-35 for the AVK-induced speed increase, aiming to support its role as a receptor for NLP-10 in vivo. Indeed, the speed increase was comparably suppressed by both the *npr-35(ok3258)* deletion and the *nlp-10* mutation (Fig. 5a). Moreover, *nlp-10; npr-35* double mutants did not exhibit an exacerbated phenotype, suggesting that NLP-10 and NPR-35 function in the same pathway as a ligand-receptor pair. The increased body bending in *flp-1* mutants following AVK photoactivation was similarly suppressed by *nlp-10* and *npr-35* single mutations, as well as by double mutation of *nlp-10* and *npr-35* (Supplementary Fig. 10a). In addition, the loss of *npr-35* completely suppressed the enhanced speed increase and the increased body bending caused by expressing NLP-10 in AVK (Fig. 5b and Supplementary Fig. 10b). The expression of NPR-35 from its own promoter rescued the compromised speed increase in *npr-35; flp-1* double mutants compared to *flp-1* mutants (Fig. 5c). In conclusion, our findings strongly suggest that NPR-35 is indeed responsible for the increase of locomotion speed and body bending, serving as the sole receptor for NLP-10, at least in this specific context.

### NPR-35 excites interneurons that promote forward locomotion

Which neurons may be the focus of action of NLP-10 through its receptor NPR-35? *npr-35* is expressed in several neurons, including premotor interneurons that regulate forward or backward locomotion (Supplementary Fig. 6c)[37]. Expression of NPR-35 in either AIY or AVB neurons, using the *ttx-3*[69] or *lgc-55B*[70] promoters, respectively, rescued the compromised speed increase in *npr-35; flp-1* double mutants (Fig. 5c). Furthermore, it rescued the abolished speed increase in *npr-35* mutants overexpressing NLP-10 in AVK (Fig. 5d). Considering that the body bending decreased upon AVK photoactivation in *npr-35* mutants expressing NPR-35 in AIY (Supplementary Fig. 10d, e), and that *flp-1* mutants increase body bending upon AVK photoactivation, it suggests that the primary target of FLP-1's inhibitory action might be the NLP-10-mediated transmission from AVK to AVB. Taken together, these findings suggest that NPR-35 functions in AVB and AIY to accelerate locomotion.

Given that AIY[71] and AVB[43,72] are known to be involved in the promotion of forward locomotion, it appears likely that NPR-35 activates these neurons in response to NLP-10. We thus recorded $Ca^{2+}$ transients in AIY before and during AVK-photoactivation (Fig. 6a–c). Though AIY was highly active before the stimulus, possibly as the immobilization startles sensory neurons innervating AIY[73–79], AVK photoactivation still significantly activated AIY in animals cultivated with ATR, but not without ATR. In *npr-35* mutants, this evoked AIY activity was more discontinuous, suggesting that NLP-10/NPR-35 signaling from AVK to AIY contributes to AIY activity (Supplementary Fig. 11a). To further assess the interaction between NPR-35 and NLP-10 and determine the G protein coupling specificity of NPR-35, we expressed NPR-35 in *Xenopus laevis* oocytes and activated it by administering NLP-10. If NPR-35 couples to $G\alpha_q$, ligand binding should activate phospholipase C and the $IP_3$ receptor, leading to $Ca^{2+}$ mobilization from the ER. In oocytes, this evokes inward currents through an intrinsic $Ca^{2+}$-activated $Cl^-$-channel[80–82]. Membrane currents were measured by means of two-electrode voltage-clamp (TEVC) recordings. NPR-35 strongly responded to each of the three synthetic mature NLP-10 peptides (NLP-10-1, 2 and 3) examined (Fig. 6d, e). Treatment with the cell-permeable $Ca^{2+}$ chelator BAPTA-AM suppressed the currents evoked by NLP-10-1 in oocytes expressing NPR-35, as well as in oocytes expressing the known $G\alpha_q$-coupled mGluR1 receptor,

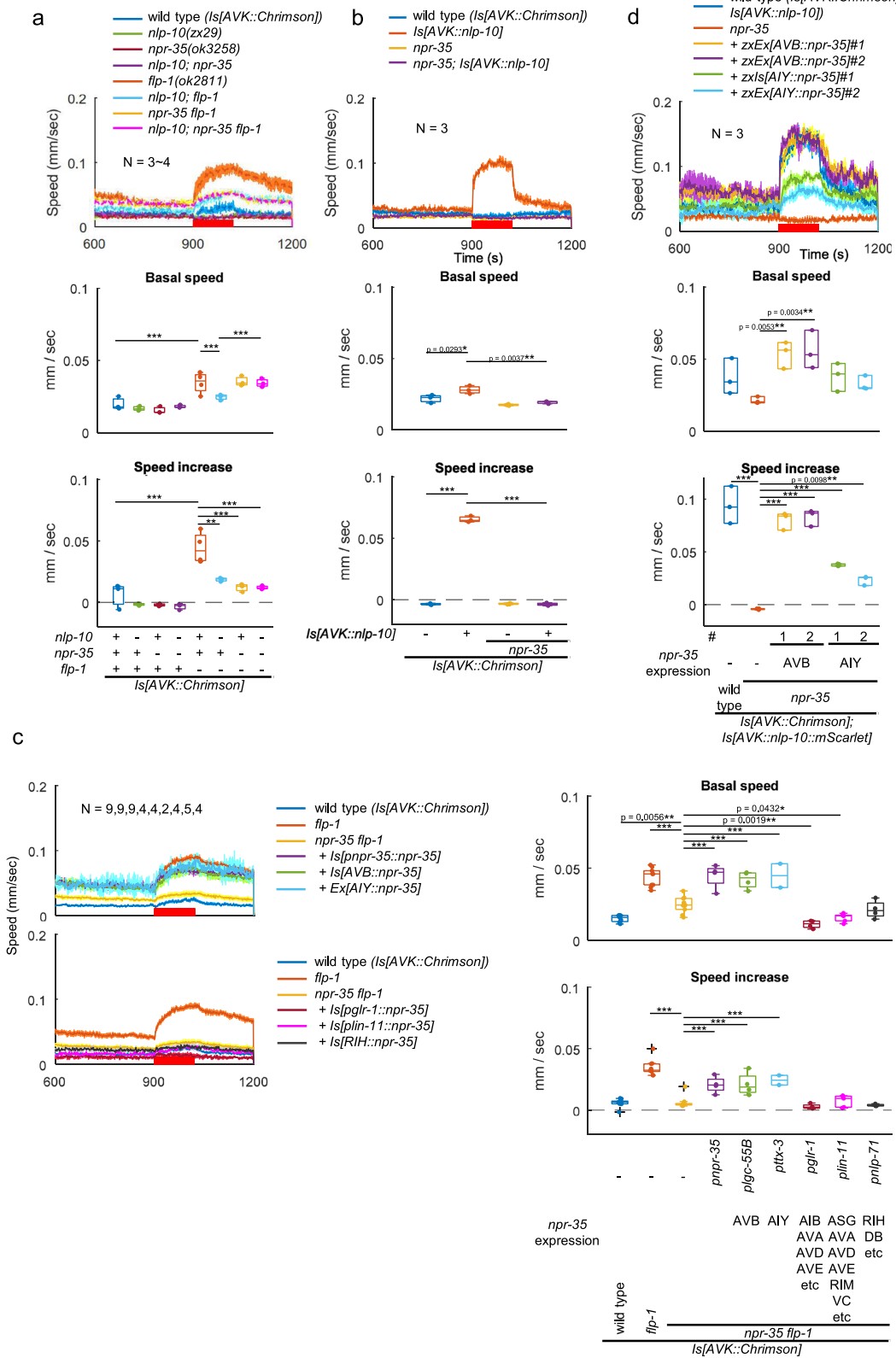

**Fig. 5 | NPR-35 mediates speed increase downstream of NLP-10. a–c** Animals of indicated genotypes expressing Chrimson in AVK were cultivated with ATR and subjected to behavioral analysis with MWT while red light was illuminated as indicated. Tukey test (**a, b**) or Dunnett test against *npr-35 flp-1* double mutants (**c**) was performed. ***$p < 0.001$. Approximately 20–120 animals were involved in each recording. **d** Animals of indicated genotypes were picked and transferred to NGM plates with ATR, cultivated overnight and subjected to behavioral analysis with

MWT. Dunnett test was performed against *npr-35* mutants. ***$p < 0.001$. Approximately 10-50 animals were involved in each recording. In timeseries plots, data are presented as mean values +/- SEM. In boxplots, the boxes extend from the Q1 to Q3, with the band inside the boxes representing the median. The whiskers extend to the smallest and largest values within 1.5 times the IQR. Source data are provided in 'Source Data 1' file.

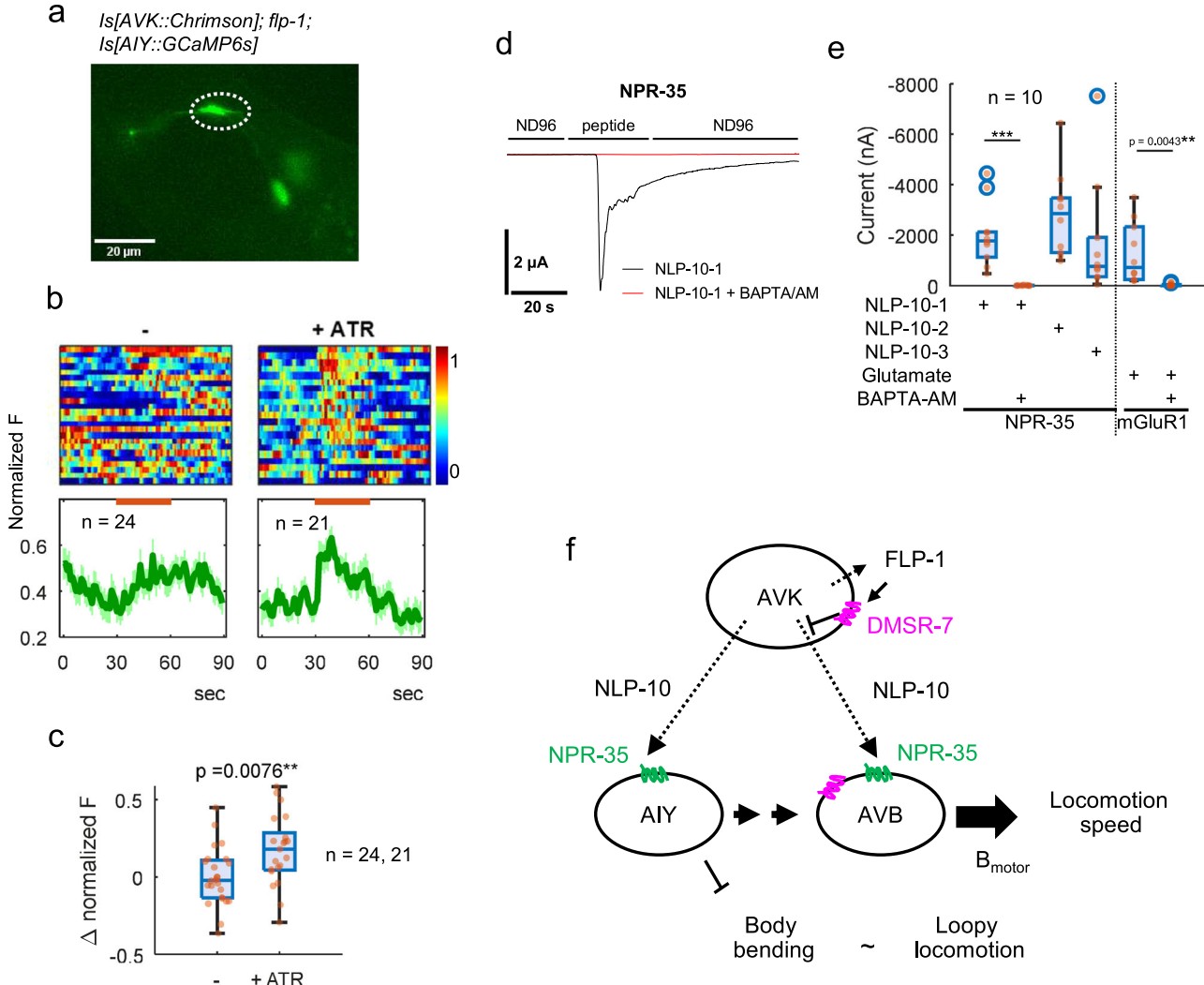

**Fig. 6 | NPR-35 is Gαq-coupled. a–c.** *flp-1(ok2811)* animals expressing Chrimson in AVK and GCaMP6s in AIY were cultivated in the presence or absence of ATR and subjected to imaging analysis. Images were acquired at 1 frame per second (fps) while orange light (590 nm) was illuminated as indicated. Raw images were subjected to background subtraction, and fluorescence intensity (F) in AIY varicosities were quantified (**a**). **b** F was normalized between 0 to 1 by subtracting the minimum values followed by division by maximum values (after subtraction). Normalized F of individual recordings were plotted in heat maps. The average of the normalized F was plotted with error bars indicating SEM. **c** Difference of normalized F before (21–30 s) and after (31–40 s) illumination was plotted. *p* value is indicated (Two-tailed Welch test). *n* represents the number of recordings. **d** Two-electrode voltage-clamp (TEVC) recordings from untreated (black) and BAPTA-AM treated (red) *Xenopus laevis* oocytes expressing NPR-35 and perfused with 100 nM NLP-10-1 in ND96 bathing solution. **e** Oocytes expressing either NPR-35 or mGluR1a were or were not treated with BAPTA-AM and perfused with 100 nM NLP-10 peptides or 100 nM glutamate, respectively, as indicated. Activation current was calculated by subtracting baseline from the maximum current and plotted. Two-tailed Welch test was performed. ***$p < 0.001$. *n* represents the number of recordings. **f** AVK-derived FLP-1 slows down locomotion speed and reduces body bending angles by suppressing NLP-10 release from AVK through autocrine feedback via the DMSR-7 receptor in AVK. NLP-10 accelerates locomotion speed by activating AVB and AIY, promoting forward locomotion. The directionality of the locomotion is dependent on the concurrent presence of FLP-1. FLP-1/DMSR-7 signaling can also affect AVB to slow down locomotion. In boxplots, the boxes extend from the Q1 to Q3, with the band inside the boxes representing the median. The whiskers extend to the smallest and largest values within 1.5 times the IQR. Source data are provided in 'Source Data 1' file.

suggesting that NPR-35 is also Gαq-coupled. Consistently, AIY activity induced by AVK photoactivation was mostly abolished by mutation of *egl-30*, encoding Gαq (Supplementary Fig. 11b). In summary, NLP-10 released from AVK likely activates AIY and AVB through activation of the Gαq-coupled GPCR NPR-35, promoting an increase in locomotion speed (Fig. 6f).

**NLP-10/NPR-35 signaling facilitates the tap response and escape from noxious blue light exposure**

We discovered the role of NLP-10 and NPR-35 as locomotion accelerators during AVK photoactivation. To determine whether they are required for natural locomotion control, we initially addressed the

tapping response, investigating whether NLP-10 antagonizes FLP-1 in a physiological context. The speed increase in response to tapping was reduced in *nlp-10* compared to wild type (Fig. 7a). Similarly, *npr-35* mutants showed reduced responses, and *nlp-10; npr-35* double mutants did not show any additive effect compared to respective single mutants. These mutations also mitigated the enhanced response of *flp-1* mutants. Our findings suggest that during natural speed increase, NLP-10/NPR-35 signaling is required and antagonizes FLP-1 effects, mirroring the observations during AVK photoactivation.

The compromised speed increase observed in *nlp-10* mutants was rescued by expressing NLP-10 either in AVK using the *twk-47* promoter or in non-AVK neurons, some of which natively express NLP-10, using

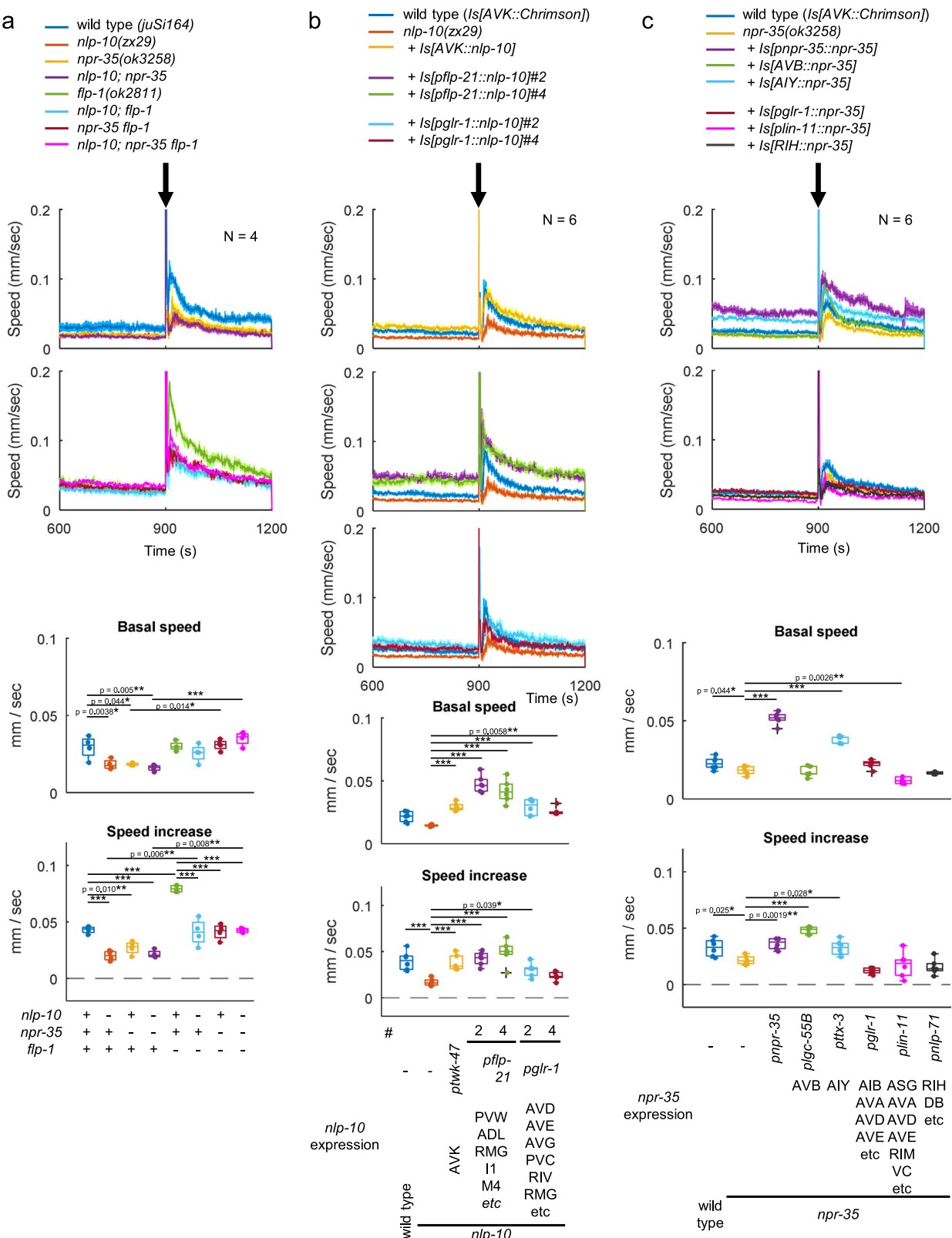

the *flp-21* promoter (Fig. 7b). This outcome indicates that NLP-10 derived from multiple neurons contributes to the tap response, in contrast to our findings for AVK photoactivation (Fig. 3m–o), posing the question if FLP-1 derived from AVK can also suppress the release of NLP-10 from these other neurons. The impaired speed increase in *npr-35* mutants was rescued by expressing NPR-35 from its own promoter, in AVB or in AIY (Fig. 7c), suggesting that NLP-10/NPR-35 signaling

activates these interneurons to accelerate locomotion, as was the case for AVK photoactivation (Fig. 5c, d).

We next investigated the role of NLP-10/NPR-35 signaling in the escape response to noxious blue light. As previously observed (Supplementary Fig. 1l–q and 2e, f), blue light illumination increases locomotion speed. When exposed to a strong and continuous blue light stimulus (470 nm, 1.3 mW/mm², 30 s), sufficient to saturate the speed

**Fig. 7 | NLP-10/NPR-35 signal facilitates tap response. a** Locomotion of animals of indicated genotypes was tracked with MWT while NGM plates were tapped. CZ20310 *juSi164[mex-Sp::His-72::miniSOG + Cbr-unc-119(+)] unc-119(ed3)* was used as wild type since the genomic loci of *juSi164* and *nlp-10(zx29)* were too close to be segregated. Tukey test was performed. ***$p < 0.001$. Approximately 20–100 animals were involved in each recording. **b** *nlp-10(zx29)* mutants expressing NLP-10 in indicated neurons, along with wild type and *nlp-10* mutant animals, were analyzed as described in **a**. Dunnett test was performed against *nlp-10* mutants. ***$p < 0.001$.

Approximately 40–100 animals were involved in each recording. **c** *npr-35(ok3258)* animals expressing NPR-35 in indicated neurons, along with wild type and *npr-35* mutant animals, were analyzed as described in **a**. Dunnett test was performed against *npr-35* mutants. ***$p < 0.001$. Approximately 20–100 animals were involved in each recording. In timeseries plots, data are presented as mean values +/- SEM. In boxplots, the boxes extend from the Q1 to Q3, with the band inside the boxes representing the median. The whiskers extend to the smallest and largest values within 1.5 times the IQR. Source data are provided in 'Source Data 1' file.

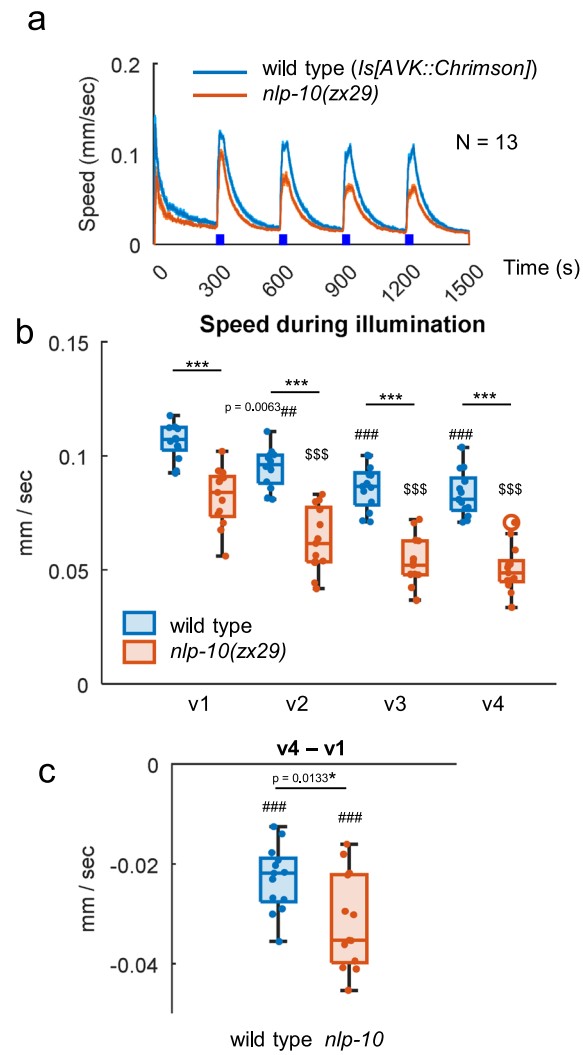

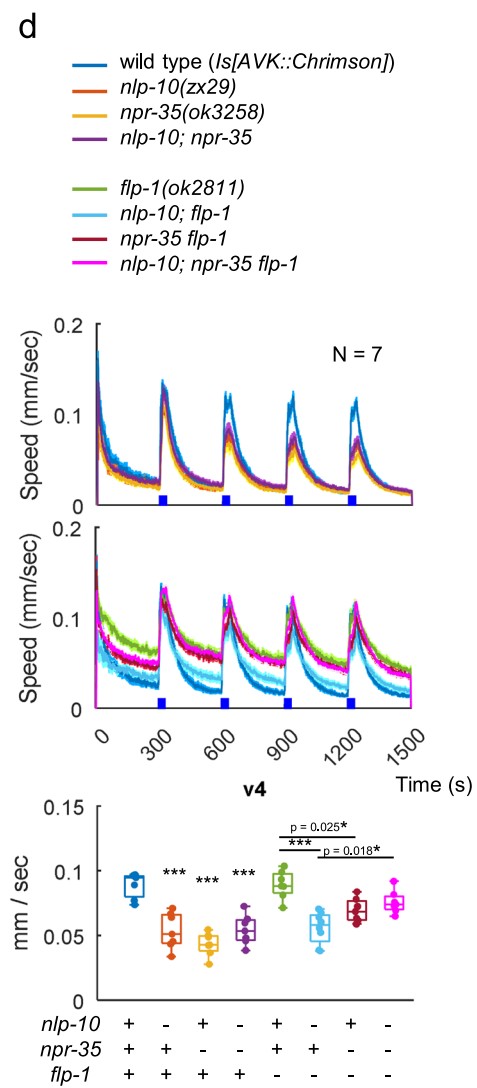

**Fig. 8 | NLP-10 suppresses habituation to repeated blue light stimuli. a–c** Wild type (*zxIs153 juIs164*) and *nlp-10(zx29)* animals were exposed to blue light (470 nm, 1.3 mW/mm²) for 30 s four times, as indicated, while their locomotion was monitored by MWT. **a** Locomotion speed was plotted over time (seconds). **b** Speed during 1st (t = 301:330) to 4th (t = 1201:1230) illumination ($v_1$ to $v_4$) and (**c**) the difference between the $v_1$ and $v_4$ were plotted. Despite expressing Chrimson in AVK, experiments were conducted in the absence of ATR, representing the response derived from innate mechanisms within the animals. Data points overlap with Fig. 8d and 9a. In (**b**), *** indicates $p < 0.001$ (Two-tailed Welch test), ## and ### indicate $p < 0.01$ and $p < 0.001$ (Dunnett test against $v_1$ of wild type) and $$$

indicates $p < 0.001$ (Dunnett test against $v_1$ of *nlp-10*). In (**c**), * indicates $p < 0.05$ (Two-tailed Welch test), and ### indicates $p < 0.001$ (One-sample *t*-test). Approximately 50–200 animals were involved in each recording. **d**. Animals with indicated genotypes were analyzed as in **a**. Speed during $v_4$ was plotted. Tukey test was performed. ***$p < 0.001$. Approximately 50–200 animals were involved in each recording. In timeseries plots, data are presented as mean values +/- SEM. In boxplots, the boxes extend from the Q1 to Q3, with the band inside the boxes representing the median. The whiskers extend to the smallest and largest values within 1.5 times the IQR. Source data are provided in 'Source Data 1' file.

increase, *nlp-10* mutants exhibited a slightly diminished response compared to wild type animals (Fig. 8a, b). With repeated illumination, this reduction of responsiveness in *nlp-10* mutants became progressively more pronounced. Notably, the difference between the 4th and the 1st responses was significantly larger in *nlp-10* mutants (Fig. 8c).

This is reminiscent of findings involving neuropeptide signaling by FLP-14 and its receptor FRPR-19, which maintain responsiveness to repeated nociceptive stimuli such as heat and harsh touch, or optogenetic activation of the FLP nociceptive neuron[83]. While habituation to repeated blue light stimuli at 0.05 Hz or 0.4 Hz (i.e., every 20 or

2.5 seconds, respectively) was less pronounced than habituation to repeated mechanical stimuli or photoactivation of tap-responsive neurons at the same frequencies, *nlp-10* mutants exhibited reduced responsiveness rather than enhanced habituation to these high-frequency blue light stimuli (Supplementary Fig. 12a–d). Our results suggest that NLP-10 is required for normal responses to blue light and maintaining responsiveness during repeated strong stimuli at low frequency. Importantly, the blue light avoidance was not enhanced by the *flp-1* mutation (Fig. 8d and Supplementary Fig. 1l–q), suggesting that the antagonism between FLP-1 and NLP-10 is not active during nociceptive responses.

Since NLP-10 affected blue light avoidance behavior, we examined whether NPR-35 mediates this effect within the yet-to-be-defined signaling pathway governing the behavioral response to UV and blue light, initiated by the photoreceptor LITE-1. In accordance with our earlier results on AVK-evoked speed increase and tap response, *npr-35* mutants and *nlp-10; npr-35* double mutants exhibited reduced responses to blue light similar to *nlp-10* mutants (Fig. 8d). The compromised speed increase in *nlp-10* mutants was rescued by expressing NLP-10, either in AVK using the *twk-47* promoter or in non-AVK neurons, some of which natively express NLP-10, using the *flp-21* or *glr-1* promoters (Fig. 9a). This result suggests that NLP-10 derived from multiple neurons contributes to the response to blue light, as observed for the tap response. Cells other than AVK, potentially releasing NLP-10 in this context, are neurons expressing both LITE-1 and NLP-10. Two cells satisfying these criteria are PVT and AVG[37], which might release NLP-10 in response to intracellular signaling following blue light-evoked excitation through LITE-1. Indeed, defective blue light responses in *lite-1* mutants were rescued by LITE-1 expression specifically in AVG, and furthermore, AVG photoactivation by Chrimson increased locomotion speed (Fig. 9b, c). Finally, we probed where NPR-35 may act in this context. The defect in *npr-35* mutants was rescued by expressing NPR-35 from its own promoter, as well as in AVB or in AIY (Fig. 9d). This suggests that NLP-10 activates these interneurons to accelerate locomotion, as observed for speed increase evoked by AVK photoactivation or by tapping (Figs. 5c, d and 7c). In sum, these results imply that during the physiological escape from noxious blue light, NLP-10 accelerates locomotion by activating AVB and AIY through NPR-35 (Fig. 9e).

## Discussion

Neuropeptides, serving as carriers of "wireless" communication, are released extrasynaptically and enable signaling between cells that lack physical connections by synapses[1,3]. This study unveiled antagonistic effects of two neuropeptides, NLP-10 and FLP-1, originating from AVK interneurons, on locomotion speed and body posture. NLP-10 accelerated locomotion through the $G\alpha_q$-coupled GPCR NPR-35 in AVB and AIY interneurons, promoting forward locomotion, and affected body bending, which correlates with the straightness of locomotion direction. Despite the absence of anatomically defined chemical or electrical synapses between AVK and AVB/AIY[34–36], our findings suggest direct communication mediated by NLP-10 and its receptor NPR-35. Whereas the *nlp-10* mutation was identified as a suppressor of *flp-1* hyperlocomotion, the NLP-10/NPR-35 signaling pathway was found to contribute to physiological elevation of locomotion speed in responses to mechanical and optical stimuli (elaborated below). Furthermore, AVK-derived FLP-1 promoted slow but straightened locomotion by suppressing NLP-10 secretion from AVK, through autocrine feedback mediated by the FLP-1 receptor DMSR-7 within AVK itself. DMSR-7 couples to $G\alpha_{i/o}$[2,33,65], which is largely antagonistic to $G\alpha_q$[84–86] in *C. elegans*, and its inhibition of $G\alpha_S$ is well established in mammalian in vitro assays[87]. Given that FLP-1/DMSR-7 signaling decreased the release of NLP-10 but not FLP-1, it is important to clarify which of these $G\alpha$ signaling pathways specifically affects NLP-10 release.

This hierarchical regulation of NLP-10 by FLP-1/DMSR-7 signaling exemplifies an additional layer of complexity in neuropeptide signaling, wherein cascades of multiple neuropeptides expressed in the same neuron can reciprocally influence each other. These two neuropeptides, released from AVK and subject to hierarchical regulation of their antagonistic functions, coordinate both speed and directionality of locomotion, thereby enabling more efficient responses to environmental stimuli. We presented evidence that AVK-derived FLP-1 suppresses NLP-10 secretion from the same neuron, proposing this as the most probable mechanism. Nevertheless, there could be additional contributing mechanisms. First, FLP-1 might compete for the NPR-35 receptor with NLP-10, although the evolutionary relationships of these peptide/GPCR systems suggest that they likely act in a cascade[2]. Second, since AVB also expresses DMSR-7 and its expression in AVB partially rescued the defect of *dmsr-7* mutants (Supplementary Fig. 8c), FLP-1 might directly affect AVB, alongside its primary effect on AVK via DMSR-7 and subsequent suppression of NLP-10/NPR-35 signaling. Third, the promiscuous nature of both FLP-1 and DMSR-7 suggests the possibility of a feedback loop involving multiple neurons. In this scenario, AVK-derived FLP-1 may affect neurons other than AVK, including AVB, potentially enhancing the release of neuropeptides that bind to DMSR-7 on AVK and suppress NLP-10 release.

We suggest that FLP-1 could operate in a spatially specific manner, since the enhanced speed increase of *flp-1* mutants was mitigated by FLP-1 expression in AVK but exacerbated by expression in AIY (Supplementary Fig. 8a). This spatial specificity could be due to FLP-1 signaling through autocrine feedback in AVK in this context. In contrast to FLP-1, NLP-10 might act over longer distance, akin to some mammalian (neuro)peptides such as leptin, insulin and oxytocin[7,12]. This inference is supported by the impaired response of *nlp-10* mutants to tapping and blue light being rescued by NLP-10 expression either in AVK or in non-AVK neurons (Figs. 7b and 9a). Notably, the AVK-evoked speed increase was enhanced only by NLP-10 expression in AVK but not in non-AVK neurons (Fig. 3m–o). Therefore, it appears that the precise temporal secretion of adequate amounts of NLP-10, rather than spatial specificity of the release site, triggers the observed speed increase. Considering that AVK-derived FLP-1 suppresses the response to tapping but not to blue light, and recognizing that NLP-10, derived from multiple neurons including AVK, could facilitate both responses, it is plausible that AVK-derived FLP-1 selectively inhibits the release of NLP-10 from neurons involved in the tapping response in addition to AVK.

The postulated difference in operational ranges of FLP-1 and NLP-10 aligns with FLP-1's ability to activate a variety of receptors, contrasting with NLP-10's specific activation of NPR-35[2]. Site-specific action is implied for several mammalian neuropeptides such as neuropeptide Y, dynorphin and somatostatin, which are expressed in various functionally distinct brain regions[7]. Consequently, these peptides are presumed to engage in local signaling, influencing target cells to yield context-dependent outcomes. Future work will explore from which regions of AVK neurons FLP-1 is released, whether the release sites of FLP-1 and NLP-10 are different, how far these neuropeptides can travel inside the animals' body to reach different sets of receptors on diverse target cells, and whether these release sites and operation ranges vary in different contexts such as locomotion regulation and pathogen avoidance[33].

### NLP-10/NPR-35 signaling contributes to the nociceptive blue light avoidance

*C. elegans* exhibits aversion to blue light, primarily relying on LITE-1. Our results demonstrated the contribution of NLP-10/NPR-35 signaling to this process. *lite-1* is expressed in a small number of neurons and was shown to act in at least one head sensory neuron, ASK[88]. However, since posterior blue light illumination accelerates locomotion just like

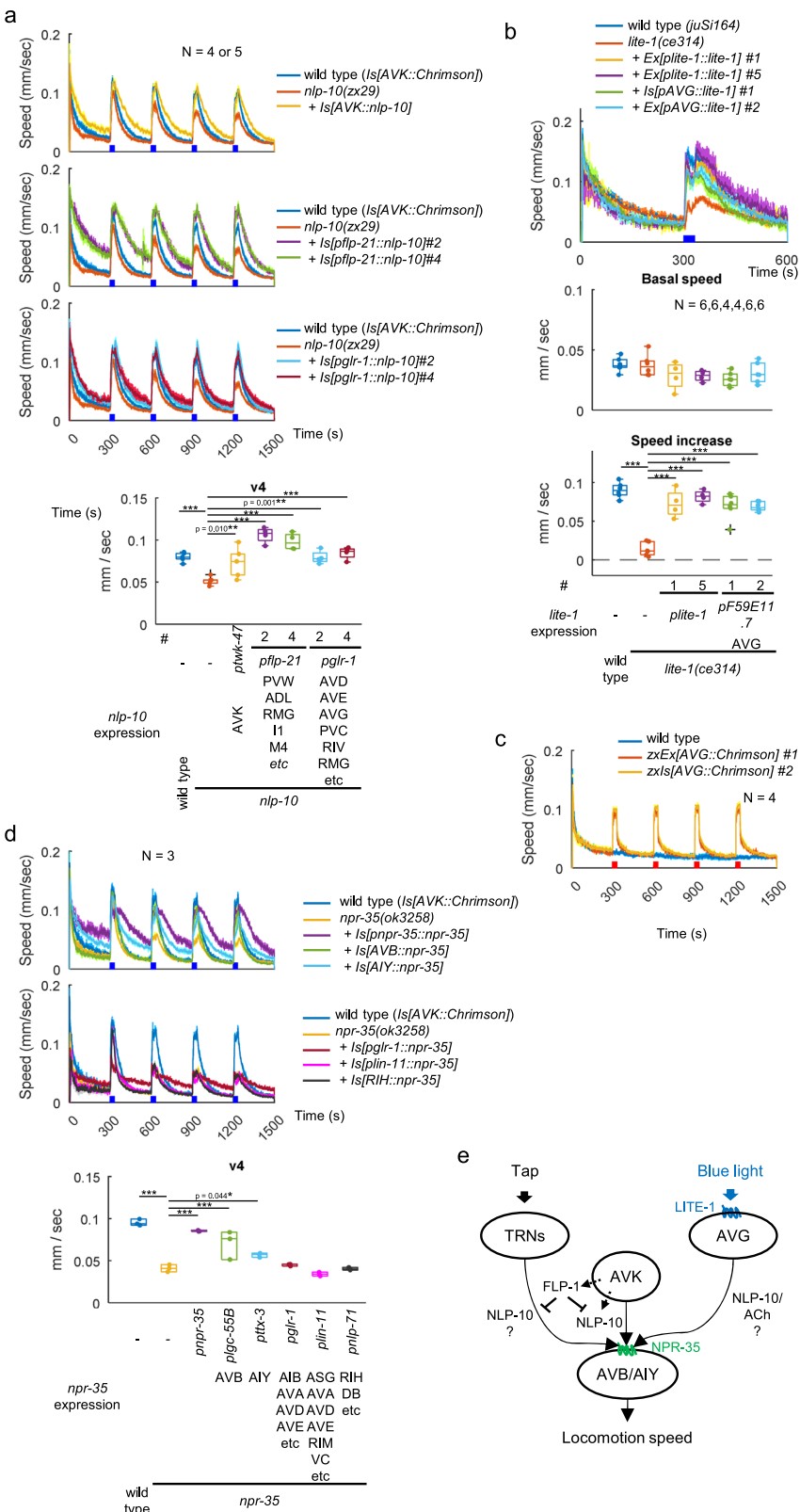

whole-body illumination[48], and ASK lacks a posterior neurite, neurons with soma or neurite in the posterior body must contribute to the response. Among *lite-1* positive neurons, AVG and PVT have neurites or cell body, respectively, in the posterior body, and also express *nlp-10*, suggesting that they may relay blue light sensation to motor circuits and sustain responsiveness by releasing NLP-10. We found that LITE-1 expression in AVG rescued defects observed in *lite-1* mutants, and that

AVG photoactivation accelerated locomotion (Fig. 9b, c). This collectively suggests that AVG is activated by blue light exposure, initiating signaling that leads to the avoidance behavior. AVG may directly activate AVB and the circuit for forward locomotion through co-transmission of ACh and NLP-10, in parallel to signals via AVK. Given the dense connectivity between PVT and AVK through gap junctions, PVT might activate AVK in response to blue light, in addition to

**Fig. 9 | NLP-10/NPR-35 signaling facilitates escape from noxious blue light stimuli. a** Animals with indicated genotypes were analyzed as in Fig. 8. Speed during $v_4$ was plotted. Dunnett test against *nlp-10* mutants was performed. ***$p < 0.001$. Approximately 50–200 animals were involved in each recording. **b** L4 larvae of indicated genotypes were picked and transferred to NGM plates one day before experiment and subjected to behavioral analysis with MWT while exposed to continuous blue light for 30 seconds, as indicated. Dunnett test was performed against *lite-1* mutants. ***$p < 0.001$. Approximately 10–40 animals were involved in each recording. **c** Animals expressing Chrimson in AVG were cultivated with ATR, and exposed to red light, as indicated. Approximately 50–150 animals were involved in each recording. **d** Animals with indicated genotypes were analyzed as described in Fig. 8. Dunnett test was performed against *npr-35* mutants.

***$p < 0.001$. Approximately 50-200 animals were involved in each recording. **e** NLP-10 derived from AVK and other neurons contributes to tap- and blue light-evoked speed increase through NPR-35 on AVB and AIY. AVG, activated by blue light via LITE-1, might stimulate AVB through the co-transmission of NLP-10 and ACh in parallel to the AVK pathway. As some touch receptor neurons (TRNs) also express *nlp-10*, AVK-derived FLP-1 may selectively suppress speed increase upon gentle mechanical stimulation by reducing NLP-10 release form these specific neurons. In timeseries plots, data are presented as mean values +/- SEM. In boxplots, the boxes extend from the Q1 to Q3, with the band inside the boxes representing the median. The whiskers extend to the smallest and largest values within 1.5 times the IQR. Source data are provided in 'Source Data 1' file.

releasing NLP-10. Future experiments will clarify the precise mechanisms by which blue light accelerates locomotion and how NLP-10 modifies this response.

### Autocrine feedback enables functional interactions between neuropeptide co-transmitters

Autocrine signaling is extensively documented in various biological processes, playing crucial roles in cancer pathogenesis[89], insulin secretion regulation[90], etc., and is important in the nervous system as well, often as a positive feedback loop. Neurotrophins such as brain-derived neurotrophic factor (BDNF) or nerve growth factor (NGF)[91] provide essential survival cues to neurons. Autocrine positive feedback involving BDNF and its receptor TrkB is documented in both pre-[92] and post-synaptic[93] cells during synaptic plasticity and axon development[94]. In specific examples, *Aplysia* demonstrates an autocrine positive feedback loop involving a BDNF analog (ApNT), the neuropeptide sensorin[95] and transforming growth factor–β (TGF-β)[96] in the presynaptic sensory neuron during synaptic plasticity consolidation. In *Drosophila*, an autocrine feedback loop by short neuropeptide F (sNPF) emerges in odorant receptor neurons, particularly under starvation conditions, induced by the upregulation of its receptor[97]. Further, oxytocin and vasopressin released from the soma and dendrites of magnocellular neurons in the hypothalamus engage in autocrine signaling through autoreceptors. This elicits retrograde feedback through endocannabinoids, inhibiting presynaptic GABAergic transmission to these neurons[98,99].

In *C. elegans*, several cases of autocrine positive feedback by neuropeptides have been suggested, which may reinforce specific cellular states. For instance, the neuropeptide pigment dispersing factor-2 (PDF-2), originating from RIM interneurons, appears to form an autocrine feedback loop relevant to multisensory decision-making[100]. Similar autocrine excitatory feedback by PDF has also been documented in circadian clock neurons in *Drosophila*[101]. Another example involves FLP-2 neuropeptides, derived from ASI sensory neurons, and the FRPR-18 receptor in ASI, suggesting the formation of an autocrine feedback loop that promotes locomotion during lethargus[102]. Last, FLP-14 neuropeptides released from *glr-1*-expressing interneurons, known for promoting reversals, might form an autocrine loop onto those neurons through the FRPR-19 receptor, particularly after photoactivation of the nociceptive neuron FLP[83].

In this study we uncovered a negative autocrine feedback loop mediated by neuropeptides. This mechanism potentially enables temporal integration of one signal before affecting another. Neuropeptide network maps predict many putative autocrine feedback motifs, consisting of neurons expressing both neuropeptides and their cognate receptors, not exclusive to *C. elegans*[1] but also in the rodent[103] nervous system. Therefore, autocrine peptidergic signaling may facilitate mutual regulation between neuropeptide co-transmitters, optimizing brain function across species. Future work will define the

mechanistic aspects of this distinct and specific signaling involving multiple neuropeptides.

## Methods
### Experimental model and subject details

*C. elegans* strains were cultivated on nematode growth medium (NGM) plates supplemented with 200 μg/ml of streptomycin seeded with *E. Coli* OP50-1 strain (Caenorhabditis Genetics Center (CGC), Twin Cities, MN, USA) in a circle as described[104], unless otherwise described. N2 (Bristol) was used as the wild type strain unless otherwise indicated. All the experiments were performed on hermaphrodites. Transgenic lines were generated by injecting plasmid DNA directly into hermaphrodite gonads as described[105]. Strains used in this study were listed in Supplementary Data 1.

Transgenes were integrated into genomes by optically activating Histone-miniSOG to mutagenize genomes as described previously[106]. Blue light was illuminated from Power-LED-Module MinoStar (2.37 W, 36 lm, 30°, Signal Construct GmbH, Niefern-Oeschelbronn, Germany) powered by LCM-40 constant current LED driver (Mean Well, New Taipei, Taiwan) and regulated at 3 Hz by Arduino Duemilanove (Arduino, Turin, Italy) and CMX100D10 solid state relay (Sensata technologies, Attleboro, MA, USA). Light intensity was 1.6 mW/mm$^2$ during continuous illumination.

*nlp-49(zx25), nlp-49(zx26), nlp-49(zx27), nlp-10(zx28)* and *nlp-10(zx29)* deletion alleles were generated by CRISPR/Cas9. sgRNA sequences were selected as described previously[107,108] and using CHOPCHOP[109]. Expression of Cas9 was optimized as described[110].

For AVK-specific feeding RNAi, *zxSi9[Cbr unc-119(+); ptwk-47::rde-1:SL2:sid-1]; rde-1(ne300)* strain, which expresses *sid-1* and *rde-1* in AVK of RNAi deficient *rde-1(ne300)* mutants[111], was generated by Mos1-mediated Single Copy Insertion (MosSCI)[112]. Expression of Mos transposase was optimized as described[110].

Defolliculated *Xenopus laevis* oocytes were purchased from EcoCyte Bioscience and maintained in ND96 solution (in mM: 96 NaCl, 1 MgCl$_2$, 5 HEPES, 1.8 CaCl$_2$, 2 KCl) at 4 °C until RNA injection.

### Plasmids

DN UNC-1 was a gift from Cori Bargmann.

pCFJ2474 pEXP[Psmu-2 | cas9(PATCs) | gpd-2 tagRFP-T(myr, patcs) smu-1 UTR] and pCFJ2475 pEXP[Psmu-2 | mosase(PATCs) | gpd-2 tagRFP-T(myr, patcs) smu-1 UTR] were gifts from Christian Froekjaer-Jensen (Addgene plasmid #159816 and #159813).

To express NPR-35 in Xenopus Laevis oocytes, the cDNA sequence of *npr-35* was amplified by PCR using Q5 High-Fidelity DNA Polymerase (New England Biolabs (NEB), Ipswich, MA, USA) from cDNA of mixed-stage populations of wild-type *C. elegans* and inserted into the KSM vector, which contains Xenopus β-globin UTR regions and a T3 promoter, using HiFi assembly (NEB). Plasmid for mouse mGluR1 was prepared from mouse cDNA.

Other plasmids used in this study are listed in Supplementary Data 1. Details regarding the plasmid constructs including sequences and used primers can be found in SnapGene files uploaded to figshare.

Primers were purchased from Eurofins Genomics (Ebersberg, Germany) or Microsynth (Balgach, Switzerland).

## RNAi

Some HT115 bacteria clones expressing double strand RNA for target mRNA were recovered from the Ahringer library[113], and the sequences were confirmed. For those genes not covered in the original version of the Ahringer library, coding regions of the genomic sequence were cloned into PstI-HindIII site of L4440 vector, and the resulting plasmids were used for transformation of HCT15 cells. Used RNAi species were listed in Supplementary Data 1.

HT115 bacteria were cultured and seeded on NGM plate containing 50 mg/ml ampicillin. For AVK-specific feeding RNAi, ZX3159 and ZX3154 strains were cultured on the NGM plates seeded with HT115 bacteria with supplementation of 1 mM IPTG and 100 μM ATR for 3 days and subjected to behavioral or imaging analysis.

## Locomotion analysis

L4 or adult animals were allowed to self-fertilize on NGM plates for 4 or 3 days at 25 °C and subjected to analysis of locomotion on food unless otherwise described. 25 °C was used because the effect of AVK photoactivation was more pronounced compared to when the cultivation temperature was 20 °C (Supplementary Fig. 3e). For optogenetic experiments, NGM plates were seeded with bacteria supplemented with 100 μM ATR. Animal locomotion was measured by Multi-Worm Tracker (MWT)[42] equipped with a solenoid tapper and Dalsa Falcon 4M30 camera (Teledyne DALSA, Waterloo, Canada) with Optical Cast Plastic IR Longpass Filter (Edmund Optics, Barrington, NJ, USA) on the lens. An infrared back light was custom made with WEPIR3-S1 IR Power LED Star infrared (850 nm) 3 W (Winger Electronics GmbH & Co. KG, Dessau, Germany) powered by LCM-40 constant current LED driver and regulated with a potentiometer (Vishay Intertechnology, Inc., Malvern, PA, USA). For optogenetics and stimuli, blue and red light was illuminated from LED modules ALUSTAR (3 W, 30°, 470 or 623 nm, 30 or 86.5 lm, respectively, Ledxon GmbH, Geisenhausen, Germany), powered by LCM-40 and regulated by Arduino Uno compatible board (Joy-IT, Neukirchen-Vluyn, Germany) with custom scripts. Animal locomotion was mostly measured on bacterial lawn except for Supplementary Fig. 2g–j.

The acquired data underwent analysis through the Choreography software, which is integrated with the MWT. This software yielded output values for 'Speed' and 'body bending'. To compute 'Straightness', the following formula was applied, using a metric derived from the 'crab' values (representing speed perpendicular to body orientation) obtained from Choreography:

$$straightness = \frac{\sqrt{speed^2 - crab^2}}{speed} \quad (1)$$

The data was subsequently visualized using custom scripts in MATLAB (Mathworks, Natick, MA, USA). In timeseries plots, data are presented as mean values +/- standard error of the mean (SEM).

## Imaging analyses

Zeiss Observer Z1 (Carl Zeiss, Oberkochen, Germany) equipped with Kinetix22 (Teledyne Photometrics, Tucson, AZ, USA) and Prior Lumen LEDs (Prior Scientific, Cambridge, UK) was regulated with μManager. Illumination was synchronized for 2-color timelapse imaging with Z-stack by the Arduino Uno compatible board with AOTFcontroller firmware (https://github.com/micro-manager/mmCoreAndDevices/blob/main/DeviceAdapters/Arduino/AOTFcontroller/AOTFcontroller.ino).

Confocal microscopes SP8 (Leica Microsystems, Wetzlar, Germany) was also used.

## Coelomocyte (CC) uptake assay

Release of neuropeptides were analyzed by quantifying fluorescence of mScarlet fused to propeptide, co-released and taken up by coelomocytes basically as described before[58]. Briefly, L4 larvae of animals expressing mScarlet fused to FLP-1 or NLP-10 propeptides were mounted on slide glasses with 50 mg/ml tetramisole (Sigma-Aldrich, Burlington, MA, USA) and Polybead® Microspheres 1.00 μm (Polysciences, Warrington, PA, USA). Images were taken with Zeiss Observer Z1 with 10x objective with excitation light at 590 nm and were subjected to background subtraction and particle analysis with Fiji.

## Calcium imaging

Animals expressing GCaMP6s were mounted on slide glasses with tetramizole and Polybead® Microspheres 1.00 μm. Images were taken with Zeiss Observer Z1 with 40x objective with excitation light at 460 nm with illumination of 590 nm light to activate Chrimson. Illumination pattern was generated by the Arduino Uno compatible board with a custom script. Images were subjected to background subtraction and quantification with Fiji.

## Two-electrode voltage clamp (TEVC) recording

5′-capped cRNA was synthesized in vitro using the T3 mMessage mMachine transcription kit (Thermo Fischer Scientific, Waltham, MA, USA) using KSM plasmids linearised with NotI as templates. cRNA was finally purified with the GeneJET RNA purification kit (Thermo Fischer Scientific). *Xenopus* oocytes were placed individually into V-bottom 96 well plates and injected with RNA using the Roboinject system (Multi Channel Systems GmbH, Reutlingen, Germany). Each oocyte was injected with a total of 10 ng of the cRNA and incubated for 2 days in ND96 at 16 °C until recording.

TEVC recordings were performed using the Robocyte2 recording system (Multi Channel Systems). Recording electrodes (Multichannel systems) normally had a resistance of 0.7–2 MΩ. The pipettes were filled with electrode solution prior to their use (1.5 M KCl and 1 M acetic acid). Some oocytes were incubated at 16 °C for 3 h in a solution of BAPTA-AM (Sigma-Aldrich) at 10 μM in ND96. Oocytes were then clamped at −80 mV, and current was continuously recorded at 500 Hz while flushing ND96 for 20 s, the peptide solutions at 100 nM in ND96 for 30 s and again ND96 to rinse.

## Statistical analysis

In boxplots, the boxes extend from the first quartile (Q1) to third quartile (Q3), with the band inside the boxes representing the median. The whiskers extend to the smallest and largest values within 1.5 times the inter-quartile range (IQR), where IQR is the difference between Q3 and Q1. Welch test was used for statistical test for two groups unless otherwise indicated. For multiple-comparison tests, one-way analyses of variance (ANOVAs) were performed, followed by Tukey-Kramer test or Dunnett test as indicated in each figure legend. Statistical tests used were all two-tailed and done by MATLAB. *, ** and *** indicate $p < 0.05$, $p < 0.01$ and $p < 0.001$, respectively

## Declaration of generative AI and AI-assisted technologies in the writing process

During the preparation of this work, the authors used ChatGPT 3.5 and later versions to improve language and readability. Following its use, the authors carefully reviewed and edited the content to ensure accuracy and take full responsibility for the final version of the manuscript.

## Reporting summary

Further information on research design is available in the Nature Portfolio Reporting Summary linked to this article.

## Data availability

Choreography output, image quantification data, and plasmid sequences, including their generation histories, are available in the figshare repository (https://doi.org/10.6084/m9.figshare.26513092). Due to their large size, the raw MWT outputs are available upon request. Source data are provided with this paper. All strains and plasmids generated in this study are available upon request. Source data are provided with this paper.

## Code availability

Custom codes used in this study are available at Zenodo (https://doi.org/10.5281/zenodo.13958554).

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

## Acknowledgements

We extend our gratitude for the invaluable technical assistance provided by lab members, with special thanks to Franziska Baum-bach, Hans-Werner Müller and Katharina Kuhlmeier. We appreciate the provision of some strains by the *Caenorhabditis* Genetics Center (CGC), funded by the NIH Office of Research Infrastructure Programs (P40 OD010440), and the Japanese National BioResource Project, nematode *C. elegans*. Special thanks to Prof. Oliver Hobert for critically reading the manuscript, providing valuable feedback and supplying a strain. We also acknowledge Dr. Jiajie Shao for his critical review. We are grateful to Prof. Mike Heilemann and Dr. Ashwin Balakrishnan for support with confocal microscopy and access to instrumentation. This work was funded by Goethe Uni-versity, the Deutsche Forschungsgemeinschaft (DFG, grant GO 1011/18-1, to A.G.), the European Research Council (ERC, 950328, to I.B.) and KU Leuven Research Council (C16/19/003, to I.B.).

## Author contributions

I.A. designed and performed experiments, wrote and edited the manu-script, and supervised E.D., S.B. and E.B. L.G. designed and performed experiments and wrote the manuscript. E.D., S.B. and E.B. performed experiments. I.B. designed experiments, edited the manuscript, super-vised, and secured funding. A.G. designed experiments, edited the manuscript, supervised, and secured funding.

## Funding

## Competing interests

The authors declare no competing interests.
