## [Peer Review File · Nature Communications]

REVIEWER COMMENTS

Reviewer #1 (Remarks to the Author):

This manuscript reports the identification of a negative autocrine feedback loop that controls locomotion speed in *C. elegans*. This regulation is mediated by two neuropeptides, FLP-1 and NLP-10, released from the same interneuron pair AVKL/R. The authors demonstrated that AVK-derived FLP-1 downregulated NLP-10 release from AVK by activating its receptor DMSR-7 present in the same neuron pair. And NLP-10 functions to promote movement by further activating its receptor NPR-35 on downstream interneurons. The authors then investigated how this regulation applies to natural locomotion control and found that while NLP-10/NPR-35 signaling are important for response to mechanical and noxious light stimuli, antagonism by FLP-1 is only involved in response to mechanical stimuli. This is an interesting paper that expands knowledge of neuropeptide signaling in regulating motor programs, and that establishes an antagonistic relationship between two peptides that may be released from the same neuron. Weaknesses lie in the lack of direct evidence showing that the release of NLP-10 or FLP-1 correlates with the behaviors analyzed, limiting support for the proposed negative autocrine feedback model in regulating behavior.

Major:

1. The authors propose a model that increased nlp-10 release drives locomotion in response to a number of stimuli and that this release is inhibited by FLP-1 signaling. One major prediction of this model is that these stimuli should increase NLP-10 and/or decrease FLP-1 release from AVK. The authors examined NLP-10::mScarlet secretion in flp-1 mutants, but they did not report whether there are any changes in NLP-10 or FLP-1 secretion following any of the AVK activation protocols. These experiments would lend direct support to the model. It may be difficult to observe changes in steady state coelomocyte fluorescence since these are acute stimuli. Perhaps prolonged stimulation may be needed to observe changes.
2. The stimuli used throughout for flp-1 signaling, plate tapping and AVK activation by Chrimson, are neither very physiological nor sustained. Swimming is reported to cause an increased state of excitatory motor output, and may thus represent a more physiologically relevant and sustained stimulus than the ones used here. Is thrashing rate regulated by flp-1/nlp-10 signaling and are FLP-1/NLP-10 secretion altered by swimming?
3. Fig 3: Locomotion data collected from flp-1(ok2811); zxls153[AVK::Chrimson] was shown in Figure 3A to 3D and other figures multiple times. In some figures, the increased speed was around 0.1 mm/sec, while in others, the speed was 0.05 mm/sec or lower. Why this difference? Were all the genotypes treated with the same condition?
4. Line 162 "AVK-photoactivation slightly accelerated locomotion (Figure 2A i-iii), consistent with AVK activity correlating with locomotion speed". Why do controls animals respond to AVK activation in Fig 2A but not in Fig 3?

5. Fig S7Bi: Authors conclude that mutations in *unc-31* reduce NLP-10 secretion based on a reduction in the in the *coel/soma* ratio. However, there is no change in *coelomocyte* fluorescence in *unc-31* mutants compared to wild type controls. Using the *soma* fluorescence to generate a ratio does not seem appropriate here since *soma* fluorescence of NLP-10::mScarlet could be altered by *unc-31* mutations in ways that are hard to predict. A better control for any effects *unc-31* mutations might have on transgene expression would be to normalize *coelomocyte* fluorescence to the expression levels of an independently generated AVK::mScarlet reporter in AVK. Without additional controls, it is not justified to conclude that *unc-31* promotes NLP-10 secretion from AVK.

6. Fig S7Bi: A potential explanation for why *unc-31* mutations do not decrease NLP-10 *coelomocyte* fluorescence is that NLP-10 release or *nlp-10* expression is upregulated when FLP-1 secretion is inhibited by *unc-31* mutations. Can *unc-31* mutations reduce NLP-10::mScarlet *coelomocyte* fluorescence in a *flp-1* mutant background?

7. Fig 4A: The authors conclude that there is an increase in NLP-10 secretion in *flp-1* mutants based on an increase in *coelomocyte* fluorescence of NLP-10::mScarlet, but this raises a similar concern as (5) above. The authors should conduct controls to address whether the increase in *coelomocyte* fluorescence in *flp-1* mutants arises from increases in transgene expression and/or increased uptake by *coelomocytes*. This is critical since the major conclusion of the paper (that FLP-1 negatively regulates the release of NLP-10) hinges on this observation. In addition, presenting the *coel/soma* ratio of NLP-10::mScarlet does not seem relevant here for the reasons outlined above, and should be omitted.

8. Line 247: "FLP-1 could suppress the speed increase either 1) through AVK-intrinsic effects on the expression or release of NLP-10...". Much of the antagonism of *nlp-10* by *flp-1* signaling could be explained if *flp-1* negatively regulates the expression of *nlp-10* instead of or in addition to its secretion, but the authors did not address whether *flp-1* regulates *nlp-10* expression. This should be experimentally addressed.

9. Fig, 5E: If NLP-10/NPR-35 signaling is responsible for activation of AIY by AVK, then mutations in *nlp-35* should block the increase GCaMP activation. Is this the case?

10. Fig. 5F: The authors show indirectly that Gaq may be a NPR-35 effector. Does knockdown of any of the known Gaq mutants (e.g., *egl-30*) in AIY block *nlp-10* signaling?

Minor:

1. Figures throughout hard to follow. The colors in the key do not seem to match the colors in the traces (maybe because the error bars are different shades of the colors) and some traces are obscured by other traces so it is challenging to know which genotypes are which in the traces. In addition, the nomenclature throughout seems to be inconsistent and unnecessarily complicated. The use of array names and promoter gene names in the figures are not helpful for understanding the experiments and are distracting. Array names and their identities could be moved to the figure legends and/or methods for those who are interested in these details. Promoter gene names (eg *ttx-3*) in the figure panels could be replaced with the cell they are meant to express in (eg. AIY).

2. Fig 3: The replicates presented do not add to understanding the paper. Replicate data could be moved to supplement to increase clarity in data presentation.
3. Line 341: “AVB or AIY” should read “AIY or AVB”
4. Page 28, line 491, “weather” should be “whether”
5. Throughout, please label the x axis in tracking graphs.

Reviewer #2 (Remarks to the Author):

Comments:

The manuscript by Aoki et al. presents a host of experiments that explore the regulation of functionally antagonistic neuropeptides, FLP-1 and NLP-10, co-expressed in a pair of *C. elegans* neurons AVK, on locomotion speed and gait. The authors demonstrate that NLP-10 speeds up locomotion by activating Gαq-coupled GPCR NPR-35 in AVB and AIY, leading to the activation of these premotor interneurons responsible for forward locomotion. Conversely, they show that FLP-1 suppresses locomotion through autocrine feedback that inhibits the release of NLP-10 via the FLP-1 receptor DMSR-7 in AVK neurons. Besides, the authors investigated the roles of FLP-1, NLP-10, and the unraveled associated circuit in behavioral responses to mechanical and noxious blue light stimuli, expanding our knowledge of how sensory inputs are translated into motor outputs through neuropeptide signaling pathways. The study's use of genetic manipulation, optogenetics, and behavioral assays is both rigorous and technically sound. Their findings reveal a neural signaling motif by which neuropeptidergic co-transmission and autocrine negative feedback confer sensorimotor functions.

Overall, I think this is a very interesting story, offering novel findings and significant insights to the field regarding neural circuit modulation and paving the way for future studies that may further dissect these complex regulatory networks. There are a couple of places where I think the authors would need revisions and clarification on some technical issues:

Points that need to be addressed or improved:

1. In Fig. 1A, it's nice to have a schematic figure for demonstrating the behavioral assay and visualizing the locomotion data (worm trajectories). However, I think the way of presenting the behavioral data in Fig. 1A could be improved; now the image is a bit too info-intensive (probably because of the color code). It's now difficult for the reader to see how each individual worm is doing during the experiment, except for appreciating the overall dispersiveness of the trajectories (where

the color code of time may not be necessary). Some suggestions: (1) Fewer trajectories in each image; (2) Remove color code. Instead could use colors to indicate some critical events (e.g. starting point, stimulus, etc.); (3) For each image, add a few inset images to show some representative worm postures within one of the trajectories, particularly at some critical time points (e.g. basal locomotion, before/after stimuli, recovery phase, etc.)

2. In Fig. 1B viii and ix, the authors calculate "straightness" as a metric to quantify the locomotion directionality. However, the range of this metric for the groups shown in the figures seems very small (roughly 0.99 to 1, which is only ~1% change). Can the authors explain why this metric has such a small range despite the significant differences between groups? Is it because the effects of difference in locomotion directionality are small in the first place, or because of the metric's own nature which might not properly reflect the differences in locomotion directionality between groups?

3. In all the figures where the authors show the time-series plots, could the authors explain the specific reason for choosing non-zero times as a starting point (except for Fig. 7 where they start at $t=0$)? What had happened before the time window that's being shown? Could the authors shift the time axes so that they all start at $t=0$?

4. I understand that N stands for the number of independent experiments, as the authors indicated in Fig. 1 legend. However, the authors didn't indicate how many individual animals were used for each experiment, which is important components for statistical analyses.

5. Besides, I noticed N varies quite a bit (from 2 to 38) across different groups/conditions. Can the authors explain why N varies that much across conditions? How is it determined? Also, a number of groups used $N=2$ (e.g. Figs. 1C, 3E, 4B, 7D, etc.) I'm concerned that the number of technical replicates (N) might be too low for those groups to determine variations.

6. According to Methods Section "Behavioral assays", "L4 and adult animals were allowed to self-fertilize on NGM plates for 4 or 3 days at 25 C". Two questions: (1) Why choosing 25 C as a cultivation temperature instead of other temperatures such as canonical temperature 20 C? Are the behaviors studied in this manuscript dependent on temperature? (2) Were all experiemnts done in a food-provided or food-free environment? Are there any differences in the behaviors on food vs no food?

7. The authors briefly described a number of functionalities of AVK and the neuropeptides it expresses in various behaviors reported in the literature. However, there are a few particularly relevant contributions that the authors didn't discuss in relation to their own contribution: (1) Line

79, regarding AVK's ability to integrate sensory inputs via dopamine signaling, consider ref PMC10193984 which is on AVK integrating proprioceptive input via dopamine signaling. (2) Line 84, consider ref PMC4854516 which investigated the correlation between the head bending angle and locomotion speed and efficiency.

8. In Fig. 5E, can the authors indicate the number of animals used for calcium imaging analysis?

9. Line 491. "weather" should be "whether".

Reviewer #3 (Remarks to the Author):

This paper is an interesting exploration of how a cell that expresses multiple neurotransmitters, which have antagonistic actions, modulate behavior. The AVK cell in *C. elegans* expresses multiple neuropeptides. Previous work has shown that FLP-1 activity decreases waveform and speed, but the counterbalance to FLP-1 action is unknown. The authors now show that among the multitudes of neuropeptides expressed in AVK, NLP-10 signaling is a possible counterbalance to FLP-1 peptide activity. Optogenetic activation of AVK increases speed; this speed is enhanced when *flp-1* is knocked out, suggesting that AVK activation leads to increased speed and inhibition of FLP-1 signaling.

AVK does not appear to express any classical transmitters. In a comprehensive set of experiments, the authors tested neuropeptides expressed in AVK for their ability to suppress FLP-1 activity. NLP-10 signaling was identified as the counterbalance to FLP-1 signaling. *nlp-10* knockouts alone, however, do not appear to have many defects, although the authors should clarify this explicitly (for instance, what happens to body bends in *nlp-10* mutants). By contrast, overexpression of NLP-10 signaling increases waveform and speed. Furthermore, when AVK is activated optogenetically in *nlp-10*; *flp-1* mutants, there is suppression to varying degrees of several AVK activated, *flp-1* mutant phenotypes, suggesting that NLP-10 signaling acts downstream of FLP-1 activity. The authors also identify NPR-35 as the NLP-10 receptor.

The question then is how NLP-10 activity is suppressed by FLP-1 activity. By expressing *nlp-10* in different cells in *nlp-10* knockouts, expression of *nlp-10* specifically in AVK rescues the suppression phenotypes. The authors propose a model whereby release of FLP-1 peptides from AVK feeds back onto AVK to inhibit further NLP-10 peptide release from AVK. Overall, this paper is a thorough and exciting analysis of competing actions of neuropeptides released from the same cell.

Some comments that the authors should address:

1. As the authors indicate in their Discussion, the action of FLP-1 peptides could be a feedback from AVB/AIY back onto AVK (as opposed to an autofeedback loop). In particular, the highest levels of the FLP-1 receptor DMSR-7 is found in AVB. The authors only perform rescues of DMSR-7 in AVK, but do not report similar rescue experiments in AVB.
2. It is unclear whether the proposed feedback loop also affects levels of FLP-1 peptide release. Is FLP-1 signaling also affected by the feedback loop? If so, how does it affect the locomotory circuit?
3. The authors do not suggest how levels of NLP-10 release will be inhibited in AVK cells by the feedback loop. Perhaps the authors could elaborate on this point in the Discussion.

Minor comments:

1. Overall, when talking about neuropeptides, the neuropeptides do not cause any action per se on their own. Similarly, the authors should be more precise when talking about receptor signaling. It should be neuropeptide/receptor signaling or neuropeptide/receptor activity.
2. Is it possible to keep the colors consistent for the strains, even within the same figure. For instance, in Fig. 5A and Fig. 5C flp-1 is green or orange, respectively; similarly in Fig. 3A, flp-1 is orange, then in Fig. 3B, flp-1 is yellow, then in Fig. 3C flp-1 is orange again, etc. This continual change in color is difficult for a reader to follow the different strains.
3. Fig. S3ii: Image for AVK is unclear.
4. Fig. 1 & other figures: N=number of trials, but could the number of animals in the population (or range of numbers of animals) also be indicated in the legend (i.e., does the trial include 10 animals, 2 animals, etc). Similarly, all of the different parts of a figure should indicate the range of animals/trials.
5. Fig. 2Civ: Is the bacteria spread as a circle? Is that why flp-1 mutants are in a circle (i.e., to the edges)?
6. Lines 294, 299: DMSR-7 signaling in AVK.
7. Fig. 5G: DMSR-7 is also expressed in AVB and AIY (at early developmental stages).

Below are the point-by-point responses to the reviewers' comments.

Reviewer #1 (Remarks to the Author):

This manuscript reports the identification of a negative autocrine feedback loop that controls locomotion speed in *C. elegans*. This regulation is mediated by two neuropeptides, FLP-1 and NLP-10, released from the same interneuron pair AVKL/R. The authors demonstrated that AVK-derived FLP-1 downregulated NLP-10 release from AVK by activating its receptor DMSR-7 present in the same neuron pair. And NLP-10 functions to promote movement by further activating its receptor NPR-35 on downstream interneurons. The authors then investigated how this regulation applies to natural locomotion control and found that while NLP-10/NPR-35 signaling are important for response to mechanical and noxious light stimuli, antagonism by FLP-1 is only involved in response to mechanical stimuli. This is an interesting paper that expands knowledge of neuropeptide signaling in regulating motor programs, and that establishes an antagonistic relationship between two peptides that may be released from the same neuron. Weaknesses lie in the lack of direct evidence showing that the release of NLP-10 or FLP-1 correlates with the behaviors analyzed, limiting support for the proposed negative autocrine feedback model in regulating behavior.

Major:

1. The authors propose a model that increased nlp-10 release drives locomotion in response to a number of stimuli and that this release is inhibited by FLP-1 signaling. One major prediction of this model is that these stimuli should increase NLP-10 and/or decrease FLP-1 release from AVK. The authors examined NLP-10::mScarlet secretion in flp-1 mutants, but they did not report whether there are any changes in NLP-10 or FLP-1 secretion following any of the AVK activation protocols. These experiments would lend direct support to the model. It may be difficult to observe changes in steady state coelomocyte fluorescence since these are acute stimuli. Perhaps prolonged stimulation may be needed to observe changes.

When pulsed AVK photoactivation was repeated for 1 hour, animals expressing nlp-10::mScarlet first showed increase of locomotion speed as shown in Fig. 3d and e, however, their response gradually decreased. This was accompanied by increased FLP-1 release and slight decrease of NLP-10 release, suggesting that the increased FLP-1 release by repetitive AVK activation suppresses the NLP-10 release, leading to the suppression of speed increase (Supplementary Fig. 7d).

2. The stimuli used throughout for flp-1 signaling, plate tapping and AVK activation by Chrimson, are neither very physiological nor sustained. Swimming is reported to cause an increased state of excitatory motor output, and may thus represent a more physiologically relevant and sustained stimulus than the ones used here. Is thrashing rate regulated by flp-1/nlp-10 signaling and are FLP-1/NLP-10 secretion altered by swimming?

Plate tapping would not be sustained but probably physiological. In addition, given that AVK is known to be involved in behaviors startled by changes in oxygen concentration, presence of food, and of pathogenic bacteria, the optogenetic stimulation of AVK might mimic the repeated inputs from sensory systems.

Nevertheless, we tested swimming for *flp-1*, *nlp-10* and *nlp-10; flp-1* double mutants. Both *flp-1* and *nlp-10* mutants showed reduced swimming cycle compared to wild type, and the double mutants show additive effects (see rebuttal figure 1), suggesting that the contribution of these neuropeptides to swimming is rather co-operative, in contrast to the case of crawling on food, where they are antagonistic.

Since it seemed that the regulation of swimming is too different from that of the paradigms we tested in this work, we did not perform analysis of neuropeptide secretion after swimming.

Rebuttal figure 1: CZ20310 *juSi164[mex-5p::Hls-72::miniSOG + Cbr-unc-119(+)] unc-119(ed3)* as wild type and indicated mutants were transferred to M9 buffer, and swimming was observed for 10 min. Thrashing number (per minute) for each bin over 1 minute was plotted. Approximately 60-260 animals were involved in each recording.

3. Fig 3: Locomotion data collected from *flp-1(ok2811); zxIs153[AVK::Chrimson]* was shown in Figure 3A to 3D and other figures multiple times. In some figures, the increased speed was around 0.1 mm/sec, while in others, the speed was 0.05 mm/sec or lower. Why this difference? Were all the genotypes treated with the same condition?

In Fig. 3a, L4 transgenic animals were picked one day before the measurement. In b-e, parental animals were allowed to self-fertilize, and L4 larvae may contribute to the results to some extent, although small animals are filtered out during the data acquisition based on the pixel size of animal silhouette. This was clarified in the figure legend.

4. Line 162 “AVK-photoactivation slightly accelerated locomotion (Figure 2A i-iii), consistent with AVK activity correlating with locomotion speed”. Why do controls animals respond to AVK activation in Fig 2A but not in Fig 3?

In the previous Fig 2A, data from animals picked one day before the experiment and those self-fertilized were mixed. Therefore, we collected recent data only from those self-fertilized. In the current dataset, the wild type animals still showed a slight speed increase.

Given that cultivation temperature affects the speed increase in wild type as discussed below (Supplementary Fig. 3e), external conditions may cause the variability of the behavior. Although we keep the temperature of the incubator and that around the experiment setup as constant as possible, humidity, for example, cannot be controlled in our facility. Actually, we first thought AVK-photoactivation does not affect the locomotion speed, while watching only wild type. Nevertheless, wild type animals still responded to AVK photoactivation with the decrease of body bending a bit more reliably than with speed increase (Supplementary Fig. 5), and more importantly, the difference between wild type and *flp-1* mutants are very robust both in speed increase and the change of body bending.

5. Fig S7Bi: Authors conclude that mutations in *unc-31* reduce NLP-10 secretion based on a reduction in the in the coel/soma ratio. However, there is no change in coelomocyte fluorescence in *unc-31* mutants compared to wild type controls. Using the soma fluorescence to generate a ratio does not seem appropriate here since soma fluorescence of NLP-10::mScarlet could be altered by *unc-31* mutations in ways that are hard to predict. A better control for any effects *unc-31* mutations might have on transgene expression would be to normalize coelomocyte fluorescence to the expression levels of an independently generated AVK::mScarlet reporter in AVK. Without additional controls, it is not justified to conclude that *unc-31* promotes NLP-10 secretion from AVK.

6. Fig S7Bi: A potential explanation for why *unc-31* mutations do not decrease NLP-10 coelomocyte fluorescence is that NLP-10 release or *nlp-10* expression is upregulated when FLP-1 secretion is inhibited by *unc-31* mutations. Can *unc-31* mutations reduce NLP-10::mScarlet coelomocyte fluorescence in a *flp-1* mutant background?

(5 and 6 together) According to the reviewer's suggestion, we expressed NLP-10::mScarlet and GFP from the same *twk-47* promoter in AVK, and normalized mScarlet fluorescence levels assessed in coelomocytes by dividing with the GFP fluorescence value measured in the AVK soma. Both the net mScarlet fluorescence in coelomocytes (F_{CC1}) and the normalized value (F_{CC1}/F_{GFP}) were still not significantly different between wild type and *unc-31* single mutants. However, as the reviewer suggested in 6, in the *flp-1* background, the *unc-31* mutation remarkably reduced the release of NLP-10::mScarlet, suggesting that UNC-31 indeed contributes to the NLP-10 release, and that in *unc-31* single mutants, there would be a mixed effect of loss of UNC-31 and the reduction of FLP-1 release, which increases NLP-10 release.

7. Fig 4A: The authors conclude that there is an increase in NLP-10 secretion in *flp-1* mutants based on an increase in coelomocyte fluorescence of NLP-10::mScarlet, but this raises a similar concern as (5) above. The authors should conduct controls to address whether the increase in coelomocyte fluorescence in *flp-1* mutants arises from increases in transgene expression and/or increased uptake by coelomocytes. This is critical since the major conclusion of the paper (that

FLP-1 negatively regulates the release of NLP-10) hinges on this observation. In addition, presenting the coel/soma ratio of NLP-10::mScarlet does not seem relevant here for the reasons outlined above, and should be omitted.

As the reviewer suggested, we expressed NLP-10::mScarlet and GFP from the same *twk-47* promoter, and normalized mScarlet fluorescence in coelomocytes by dividing GFP fluorescence in AVK soma. The normalized fluorescence (F_{CC1}/F_{GFP}) was clearly increased in *flp-1* mutants in two sets of independent strains (Fig. 4a i & Supplementary Fig. 7c i,ii), demonstrating that FLP-1 suppresses NLP-10 release from AVK.

8. Line 247: “FLP-1 could suppress the speed increase either 1) through AVK-intrinsic effects on the expression or release of NLP-10...”. Much of the antagonism of *nlp-10* by *flp-1* signaling could be explained if *flp-1* negatively regulates the expression of *nlp-10* instead of or in addition to its secretion, but the authors did not address whether *flp-1* regulates *nlp-10* expression. This should be experimentally addressed.

We addressed whether *nlp-10* transcription level is affected by *flp-1* mutation, by imaging strains where GFP is expressed downstream of the endogenous *nlp-10* promoter (*nlp-10(syb3179[nlp-10::T2A::3×NLS::GFP])*). GFP fluorescence was not increased (rather decreased) in *flp-1* mutants, indicating that FLP-1 does not suppress the function of NLP-10 by decreasing its transcription (Supplementary Fig. 7b).

9. Fig, 5E: If NLP-10/NPR-35 signaling is responsible for activation of AIY by AVK, then mutations in *npr-35* should block the increase GCaMP activation. Is this the case?

We compared the response of AIY after AVK photoactivation in *flp-1* single mutants and in *flp-1 npr-35* double mutants. In the double mutants, the initial reaction was not blocked but the activation was not sustained compared to *flp-1* single mutants, suggesting that the NLP-10-NPR-35 signaling from AVK to AIY contributes to AIY activity (Supplementary Fig. 11a).

10. Fig. 5F: The authors show indirectly that Gaq may be a NPR-35 effector. Does knockdown of any of the known Gaq mutants (e.g., *egl-30*) in AIY block *nlp-10* signaling?

We compared the response of AIY after AVK photoactivation in *flp-1* single mutants and in *egl-30; flp-1* double mutants. In the double mutants, the response was mostly blocked, which is consistent with the role of Gaq EGL-30 as an effector of NPR-35 (Supplementary Fig. 11b). The more robust effect of the loss of *egl-30* compared to that of *npr-35* may be because EGL-30 can mediate the signaling of other GPCRs for neuropeptides and small molecule transmitters, which might affect overall Ca^{2+} fluctuations in AIY.

Minor:

1. Figures throughout hard to follow. The colors in the key do not seem to match the colors in the traces (maybe because the error bars are different shades of the colors) and some traces are obscured by other traces so it is challenging to know which genotypes are which in the traces. In addition, the nomenclature throughout seems to be inconsistent and unnecessarily complicated. The use of array names and promoter gene names in the figures are not helpful for understanding the experiments and are distracting. Array names and their identities could be moved to the figure legends and/or methods for those who are interested in these details. Promoter gene names (eg ttx-3) in the figure panels could be replaced with the cell they are meant to express in (eg. AIY).

In Fig. 4b, the traces were separately plotted for better visibility. In Fig. 5a, dashed lines were used instead of solid lines for better visibility.

Array names were mostly removed from figures and written down in the strain list (Supplementary Data). Promoter gene names were replaced by cell names as much as possible.

2. Fig 3: The replicates presented do not add to understanding the paper. Replicate data could be moved to supplement to increase clarity in data presentation.

Apologies, but we did not exactly understand which data the reviewer refers to. However, we found an error in the labeling of the genotype in Fig. 3b and corrected it to clearly indicate it is an RNAi experiment.

3. Line 341: “AVB or AIY” should read “AIY or AVB”

Corrected.

4. Page 28, line 491, “weather” should be “whether”

Corrected.

5. Throughout, please label the x axis in tracking graphs.

x-axes were labeled throughout.

Reviewer #2 (Remarks to the Author):

Comments:

The manuscript by Aoki et al. presents a host of experiments that explore the regulation of functionally antagonistic neuropeptides, FLP-1 and NLP-10, co-expressed in a pair of *C. elegans* neurons AVK, on locomotion speed and gait. The authors demonstrate that NLP-10 speeds up locomotion by activating Gαq-coupled GPCR NPR-35 in AVB and AIY, leading to the activation of

these premotor interneurons responsible for forward locomotion. Conversely, they show that FLP-1 suppresses locomotion through autocrine feedback that inhibits the release of NLP-10 via the FLP-1 receptor DMSR-7 in AVK neurons. Besides, the authors investigated the roles of FLP-1, NLP-10, and the unraveled associated circuit in behavioral responses to mechanical and noxious blue light stimuli, expanding our knowledge of how sensory inputs are translated into motor outputs through neuropeptide signaling pathways. The study's use of genetic manipulation, optogenetics, and behavioral assays is both rigorous and technically sound. Their findings reveal a neural signaling motif by which neuropeptidergic co-transmission and autocrine negative feedback confer sensorimotor functions.

Overall, I think this is a very interesting story, offering novel findings and significant insights to the field regarding neural circuit modulation and paving the way for future studies that may further dissect these complex regulatory networks. There are a couple of places where I think the authors would need revisions and clarification on some technical issues:

Points that need to be addressed or improved:

1. In Fig. 1A, it's nice to have a schematic figure for demonstrating the behavioral assay and visualizing the locomotion data (worm trajectories). However, I think the way of presenting the behavioral data in Fig. 1A could be improved; now the image is a bit too info-intensive (probably because of the color code). It's now difficult for the reader to see how each individual worm is doing during the experiment, except for appreciating the overall dispersiveness of the trajectories (where the color code of time may not be necessary). Some suggestions: (1) Fewer trajectories in each image; (2) Remove color code. Instead could use colors to indicate some critical events (e.g. starting point, stimulus, etc.); (3) For each image, add a few inset images to show some representative worm postures within one of the trajectories, particularly at some critical time points (e.g. basal locomotion, before/after stimuli, recovery phase, etc.)

(1) We selected examples with slightly less trajectories.

(2) Actually, the start of recording and the tapping are well represented by the current color code (**Fig. 1a,b**). Blue indicates the trajectory between the start of the recording and the tapping, and turquoise indicates the times right after the tapping. Enhanced and prolonged speed increases after tapping of *flp-1* mutants is thus well represented by longer traces starting with turquoise, and having long red tails. This point was described in the manuscript.

(3) For wild type and *flp-1* mutants, representative skeletons right after tapping were added. It was hard to distinguish the posture before and after tapping by eye.

2. In Fig. 1B viii and ix, the authors calculate "straightness" as a metric to quantify the locomotion directionality. However, the range of this metric for the groups shown in the figures seems very small (roughly 0.99 to 1, which is only ~1% change). Can the authors explain why this metric has

such a small range despite the significant differences between groups? Is it because the effects of difference in locomotion directionality are small in the first place, or because of the metric's own nature which might not properly reflect the differences in locomotion directionality between groups?

The 'crab' values (representing speed perpendicular to body orientation) are typically around 1/10 of 'speed' values, making the 'straightness' value around 0.99. This is probably because 'speed' and 'crab' are acquired as mean values within 1 second time windows, and worms predominantly move forward with a gradual change of direction (described as "weather-vane" steering strategy, Iino and Yoshida, 2009) and rare sharp directional changes (pirouettes). With a timeframe longer than 1 second, the values might be expanded to a broader range. Nevertheless, our current metric well represents loopy locomotion of *flp-1* mutants, for example, and our datasets show statistically significant difference.

3. In all the figures where the authors show the time-series plots, could the authors explain the specific reason for choosing non-zero times as a starting point (except for Fig. 7 where they start at $t=0$)? What had happened before the time window that's being shown? Could the authors shift the time axes so that they all start at $t=0$?

As seen in Fig. 9 and Supplementary Fig. 1a and 2, animals show high speed in the initial phase of recording, probably due to the mechanical stimulation when placing NGM plates on the Multi-Worm Tracker and chemical stimuli by high oxygen concentration when opening the lids of the plates. Since we aimed to compare the responses to stimuli such as AVK photoactivation, tapping or blue light illumination from the baseline among different strains, we focused on the time periods before and after stimulation.

4. I understand that N stands for the number of independent experiments, as the authors indicated in Fig. 1 legend. However, the authors didn't indicate how many individual animals were used for each experiment, which is important components for statistical analyses.

Approximate numbers of animals involved in each experiment are now indicated in the figure legends.

5. Besides, I noticed N varies quite a bit (from 2 to 38) across different groups/conditions. Can the authors explain why N varies that much across conditions? How is it determined? Also, a number of groups used $N=2$ (e.g. Figs. 1C, 3E, 4B, 7D, etc.) I'm concerned that the number of technical replicates (N) might be too low for those groups to determine variations.

Previously, in Fig. 2b, measurements for another screening unrelated to this manuscript were included. Now, only those related to the screening for neuropeptides expressed in AVK (Supplementary Fig. 4d) are included. To analyze subtle changes of 'straightness', for instance,

relatively large sample numbers were necessary, but for some mutants showing almost all-or-none differences, small sample numbers were sufficient.

Experiments in Figs. 1c, 3e, 4b, 4c, 9b, Supplementary Fig. 2b-d and 12d were repeated.

6. According to Methods Section "Behavioral assays", "L4 and adult animals were allowed to self-fertilize on NGM plates for 4 or 3 days at 25 C". Two questions: (1) Why choosing 25 C as a cultivation temperature instead of other temperatures such as canonical temperature 20 C? Are the behaviors studied in this manuscript dependent on temperature? (2) Were all experiments done in a food-provided or food-free environment? Are there any differences in the behaviors on food vs no food?

(1) The speed increase after AVK photoactivation was indeed affected by cultivation temperature. When animals were cultivated at 25°C, both wild type and *flp-1* mutants tended to show a larger speed increase (Supplementary Fig. 3e). This is why we used 25°C rather than 20°C.

(2) Most experiments were done on food except for Supplementary Fig. 2d. Locomotion on and off food is very different. For example, wild type animals show much higher basal speed and increased body bending off food compared to on food (Supplementary Fig. 1 and 2).

7. The authors briefly described a number of functionalities of AVK and the neuropeptides it expresses in various behaviors reported in the literature. However, there are a few particularly relevant contributions that the authors didn't discuss in relation to their own contribution: (1) Line 79, regarding AVK's ability to integrate sensory inputs via dopamine signaling, consider ref PMC10193984 which is on AVK integrating proprioceptive input via dopamine signaling. (2) Line 84, consider ref PMC4854516 which investigated the correlation between the head bending angle and locomotion speed and efficiency.

(1) While PMC10193984 had been referred to in the discussion about output from AVK, it was referred to again in the discussion about input to AVK again.

(2) PMC4854516 was added.

8. In Fig. 5E, can the authors indicate the number of animals used for calcium imaging analysis?

Numbers were indicated also in the current Fig. 6c in addition to 6b.

9. Line 491. "weather" should be "whether".

Thank you for spotting this, it has been corrected.

Reviewer #3 (Remarks to the Author):

This paper is an interesting exploration of how a cell that expresses multiple neurotransmitters, which have antagonistic actions, modulate behavior. The AVK cell in *C. elegans* expresses multiple neuropeptides. Previous work has shown that FLP-1 activity decreases waveform and speed, but the counterbalance to FLP-1 action is unknown. The authors now show that among the multitudes of neuropeptides expressed in AVK, NLP-10 signaling is a possible counterbalance to FLP-1 peptide activity. Optogenetic activation of AVK increases speed; this speed is enhanced when *flp-1* is knocked out, suggesting that AVK activation leads to increased speed and inhibition of FLP-1 signaling.

AVK does not appear to express any classical transmitters. In a comprehensive set of experiments, the authors tested neuropeptides expressed in AVK for their ability to suppress FLP-1 activity. NLP-10 signaling was identified as the counterbalance to FLP-1 signaling. *nlp-10* knockouts alone, however, do not appear to have many defects, although the authors should clarify this explicitly (for instance, what happens to body bends in *nlp-10* mutants). By contrast, overexpression of NLP-10 signaling increases waveform and speed. Furthermore, when AVK is activated optogenetically in *nlp-10*; *flp-1* mutants, there is suppression to varying degrees of several AVK activated, *flp-1* mutant phenotypes, suggesting that NLP-10 signaling acts downstream of FLP-1 activity. The authors also identify NPR-35 as the NLP-10 receptor.

The question then is how NLP-10 activity is suppressed by FLP-1 activity. By expressing *nlp-10* in different cells in *nlp-10* knockouts, expression of *nlp-10* specifically in AVK rescues the suppression phenotypes. The authors propose a model whereby release of FLP-1 peptides from AVK feeds back onto AVK to inhibit further NLP-10 peptide release from AVK. Overall, this paper is a thorough and exciting analysis of competing actions of neuropeptides released from the same cell.

nlp-10 single mutants showed a compromised response to tapping (**Fig. 7a,b**) and blue light (**Fig. 8 and Supplementary Fig. 12**) and an altered undulation cycle in liquid (**Rebuttal Fig. 1**). Body bending of *nlp-10* mutants was not significantly different from wild type in the dataset in Supplementary Fig. 5, but decreased throughout the analysis in Fig. 8a (see **Rebuttal Fig. 2**).

Rebuttal figure 2: Plot of the body bending angles during the analysis in Fig. 8a. Wild type and *nlp-10(zx29)* animals were exposed to blue light (470 nm, 1.3 mW/mm²) for 30 seconds four times, as indicated, while their locomotion was monitored by MWT.

Some comments that the authors should address:

1. As the authors indicate in their Discussion, the action of FLP-1 peptides could be a feedback from AVB/AIY back onto AVK (as opposed to an autofeedback loop). In particular, the highest levels of the FLP-1 receptor DMSR-7 is found in AVB. The authors only perform rescues of DMSR-7 in AVK, but do not report similar rescue experiments in AVB.

DMSR-7 expression in AVB indeed partially rescued the enhanced speed increase in *dmsr-7* mutants, and this was dependent on *flp-1* (Supplementary Fig. 8c, d). While *dmsr-7* is highly expressed also in VB motoneurons, DMSR-7 expression in these neurons did not rescue the effect of *dmsr-7* on locomotion speed but had some effect on body bending.

2. It is unclear whether the proposed feedback loop also affects levels of FLP-1 peptide release. Is FLP-1 signaling also affected by the feedback loop? If so, how does it affect the locomotory circuit?

FLP-1 release from AVK was not increased in *dmsr-7* mutants, in contrast to NLP-10 release being increased (Fig. 4a and Supplementary Fig. 7c). This indicates that the release of NLP-10 and FLP-1 is indeed separately regulated.

3. The authors do not suggest how levels of NLP-10 release will be inhibited in AVK cells by the feedback loop. Perhaps the authors could elaborate on this point in the Discussion.

DMSR-7 couples to $G\alpha_{i/o}$, which is largely antagonistic to $G\alpha_q$ in *C. elegans*, and its inhibition of $G\alpha_s$ is well established in mammalian *in vitro* assays. Although no known molecular mechanisms clearly connect $G\alpha_{i/o}$ with suppression of DCV exocytosis, given that FLP-1/DMSR-7 signaling decreased the release of NLP-10 but not FLP-1, it is important to clarify which of these $G\alpha$ signaling pathways specifically affects NLP-10 release.

We added this discussion in the 'Discussion' section.

Minor comments:

1. Overall, when talking about neuropeptides, the neuropeptides do not cause any action per se on their own. Similarly, the authors should be more precise when talking about receptor signaling. It should be neuropeptide/receptor signaling or neuropeptide/receptor activity.

We corrected as much as possible.

2. Is it possible to keep the colors consistent for the strains, even within the same figure. For instance, in Fig. 5A and Fig. 5C *flp-1* is green or orange, respectively; similarly in Fig. 3A, *flp-1* is orange, then in Fig. 3B, *flp-1* is yellow, then in Fig. 3C *flp-1* is orange again, etc. This continual change in color is difficult for a reader to follow the different strains.

Colors for *flp-1* and *npr-35 flp-1* were kept consistent between Fig. 5a and 5c, as well as between Supplementary Fig. 10a and 10c.

The color for *flp-1* was kept consistent among Fig. 2, and Supplementary Fig. 3b-d, among Fig. 3a, b and d, as well as among Supplementary Fig. 5a, b and d.

3. Fig. S3ii: Image for AVK is unclear.

Pictures of ZX3296 were renewed (Supplementary Fig. 3a ii).

4. Fig. 1 & other figures: N=number of trials, but could the number of animals in the population (or range of numbers of animals) also be indicated in the legend (i.e., does the trial include 10 animals, 2 animals, etc). Similarly, all of the different parts of a figures should indicate the range of animals/trials.

Approximate animal numbers involved in each recording were indicated in figure legends.

5. Fig. 2S3Civ: Is the bacteria spread as a circle? Is that why *flp-1* mutants are in a circle (i.e., to the edges)?

Yes. Bacterial food was seeded in a circle, and *flp-1* mutants tend to cluster at the edges. This is now indicated in the manuscript.

6. Lines 294, 299: DMSR-7 signaling in AVK.

Thank you, we corrected it.

7. Fig. 5G: DMSR-7 is also expressed in AVB and AIY (at early developmental stages).

Since DMSR-7 expression in AVB partially suppressed the enhanced speed increase after AVK photoactivation in *dmsr-7* mutants (Supplementary Fig. 8c), DMSR-7 was drawn on AVB.

REVIEWER COMMENTS

Reviewer #1 (Remarks to the Author):

the authors adequately addressed all of my concerns

Reviewer #2 (Remarks to the Author):

The authors have done a good job addressing my questions. Thanks!

Reviewer #3 (Remarks to the Author):

This manuscript explores a phenomenon of dual transmitters expressed and released from a single neuron, but the transmitters have opposing behavioral effects. In particular, the authors have focused on the release of multiple neuropeptides from a single neuron involved in locomotion in *C. elegans*. The authors follow up on work that the FLP-1 peptides have an inhibitory role in locomotion, while the NLP-10 peptides promote locomotion. This work led to an interesting model whereby release of FLP-1 peptides from the AVK neurons feedback onto AVK to regulate the release of NLP-10 peptides. The role of NLP-10 peptides was also found in pathways that did not involve FLP-1 peptides.

This manuscript has been revised from its previous version, addressing many of the problems that the reviewers highlighted. However, although still puzzled by their model, the work is interesting and begins the molecular dissection of how multiple transmitters released from the same neuron can have different effects.

1. The authors propose the release of FLP-1 peptides from AVK neurons feedback onto AVK to affect NLP-10 release from AVK. However, the data also support FLP-1 peptides released from AVK exciting AVB neurons, which then feedback onto AVK to affect NLP-10 release, but there is no mention of this alternative model in Fig. 6 legend. This possibility is also only cursorily mentioned in the Discussion (lines 383-385), but is as viable a possibility as the feedback loop that is being proposed. Alternatively, both mechanisms may be used to regulate NLP-10 release. Could the

authors please comment (and include in the Discussion) as to why the feedback loop is being favored?

Minor comments:

1. Line 71: instead of “FLP-1 is supposed...”, a better wording would be “FLP-1 is proposed...”
2. Fig. 1: Box plots should also include values for each animal.
3. Supp Fig 4 & 7: genes should be labeled properly; a picture of the coelomocytes with Nomarski should also be shown with the mScarlet (such as in Supp Fig 7b), because it is hard to visualize where the cells are within the animal.
4. Line 205: Changes in nlp-1 transcription levels in AVK or all nlp-10-expressing neurons? Fluorescence is not the best way of showing changes in transcription levels, but may be the best way in this case.
5. The expectation is that loss of npr-35 would have the same effects as loss of nlp-10; however, in Fig. 5a_{ii}, npr-35 flp-1 look more like flp-1 than nlp-10; flp-1 for basal speed. Could the authors please comment.
6. The noxious blue light experiments were inserted without much explanation to a reader, but is interesting because it shows a behavior that involves NLP-10 peptides but not FLP-1 peptides. A couple of sentences should be included to orient the reader (and explain what a satiating stimuli is).

Below are the point-by-point responses to the reviewers' comments.

Reviewer #1 (Remarks to the Author):

the authors adequately addressed all of my concerns

Reviewer #2 (Remarks to the Author):

The authors have done a good job addressing my questions. Thanks!

Reviewer #3 (Remarks to the Author):

This manuscript explores a phenomenon of dual transmitters expressed and released from a single neuron, but the transmitters have opposing behavioral effects. In particular, the authors have focused on the release of multiple neuropeptides from a single neuron involved in locomotion in *C. elegans*. The authors follow up on work that the FLP-1 peptides have an inhibitory role in locomotion, while the NLP-10 peptides promote locomotion. This work led to an interesting model whereby release of FLP-1 peptides from the AVK neurons feedback onto AVK to regulate the release of NLP-10 peptides. The role of NLP-10 peptides was also found in pathways that did not involve FLP-1 peptides.

This manuscript has been revised from its previous version, addressing many of the problems that the reviewers highlighted. However, although still puzzled by their model, the work is interesting and begins the molecular dissection of how multiple transmitters released from the same neuron can have different effects.

1. The authors propose the release of FLP-1 peptides from AVK neurons feedback onto AVK to affect NLP-10 release from AVK. However, the data also support FLP-1 peptides released from AVK exciting AVB neurons, which then feedback onto AVK to affect NLP-10 release, but there is no mention of this alternative model in Fig. 6 legend. This possibility is also only cursorily mentioned in the Discussion (lines 383-385), but is as viable a possibility as the feedback loop that is being proposed. Alternatively, both mechanisms may be used to regulate NLP-10 release. Could the authors please comment (and include in the Discussion) as to why the feedback loop is being favored?

Our interpretation of the data (Supplementary Fig. 8c-d) is that FLP-1/DMSR-7 signaling can directly suppress AVB, counteracting the enhanced NLP-10/NPR-35 signaling from AVK to AVB, which is caused by the loss of FLP-1/DMSR-7 autocrine feedback to AVK. This counteraction can prevent the speed increase. We propose that the primary effect of FLP-1/DMSR-7 signaling occurs in AVK, where it suppresses NLP-10 release. This is supported by the full rescue of the *dmsr-7* mutant when *dmsr-7* was expressed in AVK, compared to only partial rescue when expressed in AVB. While there is no direct evidence of feedback from AVB to AVK so far, the promiscuous nature of both FLP-1 and DMSR-7 suggests the possibility of a feedback loop

involving multiple neurons. In this scenario, AVK-derived FLP-1 may affect neurons other than AVK, including AVB, potentially enhancing the release of neuropeptides that bind to DMSR-7 on AVK and suppress NLP-10 release.

These discussions are now included in the description of Supplementary Fig. 8c-d in the 'Results' section, in the legend for Fig. 6f, and in the 'Discussion' section.

Minor comments:

1. Line 71: instead of "FLP-1 is supposed....", a better wording would be "FLP-1 is proposed..."

Thank you. It was corrected.

2. Fig. 1: Box plots should also include values for each animal.

We used the mean values of animal populations for analyses throughout this study, as is common in behavioral studies involving populations (e.g. chemotaxis/thermotaxis indices). While it is possible to analyze the data from individual animals, we do not believe this approach offers significant advantages over using group means, nor does it yield additional insights when focusing on metrics such as locomotion speed or body bending. Additionally, individual analysis is more time-consuming, since the Multi-worm tracker is not optimized for this purpose, making such analyses less practical. A key drawback of analyzing individual animal data is that measurements with larger sample sizes are disproportionately weighted.

3. Supp Fig 4 & 7: genes should be labeled properly; a picture of the coelomocytes with Nomarski should also be shown with the mScarlet (such as in Supp Fig 7b), because it is hard to visualize where the cells are within the animal.

To show where mScarlet is localized in animals in Supplementary Figs. 4g, 7b i and ,8a iii, we either merged bright field images with fluorescence images or outlined animals on the fluorescence images.

While We were not sure which gene labels in Supplementary Fig. 4g needed correction, we specified in the figure legend that *twk-47* promoter was used for AVK-specific expression. Additionally, we corrected the gene names in Supplementary Fig. 4d by adding hyphens between alphabets and numbers. The full genotypes of the animals used in Supplementary Fig. 7b are now included in the corresponding legend.

4. Line 205: Changes in *nlp-1* transcription levels in AVK or all *nlp-10*-expressing neurons? Fluorescence is not the best way of showing changes in transcription levels, but may be the best way in this case.

It is about the change in *nlp-10* transcription levels in AVK. This is now clearly indicated in the manuscript. Thank you for highlighting this.

We also believe that fluorescence is an effective and feasible method to demonstrate cell-specific transcription levels in this case.

5. The expectation is that loss of npr-35 would have the same effects as loss of nlp-10; however, in Fig. 5a_{ii}, npr-35 flp-1 look more like flp-1 than nlp-10; flp-1 for basal speed. Could the authors please comment.

Given that our de-orphanization (Beets et al., 2023) does not cover every neuropeptide in *C. elegans*, it is possible that other ligands for NPR-35 exist, which may influence basal speed in a manner opposite to NLP-10.

6. The noxious blue light experiments were inserted without much explanation to a reader, but is interesting because it shows a behavior that involves NLP-10 peptides but not FLP-1 peptides. A couple of sentences should be included to orient the reader (and explain what a satiating stimuli is).

Since the response to blue light is mentioned once (Line 119) and analyzed (Supplementary Figs. 1c and 2c), we briefly referred to these results in Line 330. The corresponding paragraph was revised accordingly.